# Differences in the Evolution of Clinical, Biochemical, and Hematological Indicators in Hospitalized Patients with COVID-19 According to Their Vaccination Scheme: A Cohort Study in One of the World’s Highest Hospital Mortality Populations

**DOI:** 10.3390/vaccines12010072

**Published:** 2024-01-11

**Authors:** Martha A. Mendoza-Hernandez, Jose Guzman-Esquivel, Marco A. Ramos-Rojas, Vanessa V. Santillan-Luna, Carmen A. Sanchez-Ramirez, Gustavo A. Hernandez-Fuentes, Janet Diaz-Martinez, Valery Melnikov, Fabian Rojas-Larios, Margarita L. Martinez-Fierro, Daniel Tiburcio-Jimenez, Iram P. Rodriguez-Sanchez, Osiris G. Delgado-Enciso, Ariana Cabrera-Licona, Ivan Delgado-Enciso

**Affiliations:** 1School of Medicine, University of Colima, Colima 28040, Mexico; mendoza_martha@ucol.mx (M.A.M.-H.); vvsantillanluna@gmail.com (V.V.S.-L.); carmen_sanchez@ucol.mx (C.A.S.-R.); gahfuentes@gmail.com (G.A.H.-F.); valery.melnikoff@gmail.com (V.M.); frojas@ucol.mx (F.R.-L.); leinad_dicapriano@hotmail.com (D.T.-J.); 1933osiris@gmail.com (O.G.D.-E.); 2General Hospital of Zone 1, Colima Delegation, Mexican Institute of Social Security, Villa de Álvarez, Colima 28984, Mexico; marcoz74@hotmail.com; 3Clinical Epidemiology Research Unit, Mexican Institute of Social Security, Villa de Alvarez, Colima 28984, Mexico; pepeguzman_esquivel@outlook.com; 4Cancerology State Institute, Colima State Health Services, Colima 28085, Mexico; arianacabrera267@gmail.com; 5Research Center in Minority Institutions, Robert Stempel College of Public Health, Florida International University, Miami, FL 33199, USA; jdiaz@caridad.org; 6Molecular Medicine Laboratory, Academic Unit of Human Medicine and Health Sciences, Autonomous University of Zacatecas, Zacatecas 98160, Mexico; margaritamf@uaz.edu.mx; 7Molecular and Structural Physiology Laboratory, School of Biological Sciences, Autonomous University of Nuevo Leon, San Nicolas de los Garza 66455, Mexico; iramrodriguez@gmail.com; 8Department of Dietetics and Nutrition, Robert Stempel College of Public Health and Social Work, Florida International University, Miami, FL 33199, USA

**Keywords:** COVID-19, hospitalized patients, vaccination scheme, mortality, diagnostic markers, cohort study

## Abstract

COVID-19 vaccines primarily prevent severe illnesses or hospitalization, but there is limited data on their impact during hospitalization for seriously ill patients. In a Mexican cohort with high COVID-19 mortality, a study assessed vaccination’s effects. From 2021 to 2022, 462 patients with 4455 hospital days were analyzed. The generalized multivariate linear mixed model (GENLINMIXED) with binary logistic regression link, survival analysis and ROC curves were used to identify risk factors for death. The results showed that the vaccinated individuals were almost half as likely to die (adRR = 0.54, 95% CI = 0.30–0.97, *p* = 0.041). When stratifying by vaccine, the Pfizer group (BNT162b2) had a 2.4-times lower risk of death (adRR = 0.41, 95% CI = 0.2–0.8, *p* = 0.008), while the AstraZeneca group (ChAdOx1-S) group did not significantly differ from the non-vaccinated (adRR = 1.04, 95% CI = 0.5–2.3, *p* = 0.915). The Pfizer group exhibited a higher survival, the unvaccinated showed increasing mortality, and the AstraZeneca group remained intermediate (*p* = 0.003, multigroup log-rank test). Additionally, BNT162b2-vaccinated individuals had lower values for markers, such as ferritin and D-dimer. Biochemical and hematological indicators suggested a protective effect of both types of vaccines, possibly linked to higher lymphocyte counts and lower platelet-to-lymphocyte ratio (PLR). It is imperative to highlight that these results reinforce the efficacy of COVID-19 vaccines. However, further studies are warranted for a comprehensive understanding of these findings.

## 1. Introduction

Severe acute respiratory syndrome coronavirus 2 (SARS-CoV-2) is the causative agent of coronavirus disease (COVID-19), a genuinely multisystemic illness despite its primary impact on the respiratory system [1,2]. While most infections are self-limiting or necessitate ambulatory care, 20% of symptomatic, unvaccinated adults may require hospitalization [2]. Vaccination has been shown to reduce this hospitalization rate by up to 10 times [3,4]. Regional and country-specific factors can influence the proportion of patients developing severe/critical COVID-19 and requiring hospitalization [5].

During the initial years of the pandemic, various types of COVID-19 vaccines were developed. The most widely distributed include: (1) mRNA-based vaccines such as BNT162b2 (Pfizer—BioNTech) and mRNA-1273 (Moderna—NIAID); (2) adenoviral vector-based vaccines like ChAdOx1-S (Oxford—AstraZeneca), Ad26.COV2.S (Johnson & Johnson), Gam-COVID-Vac (Gamaleya), and Ad5nCoV (CanSino); and (3) inactivated coronavirus-based vaccines like CorovaVac (Sinovac) and BBIBP-CorV (Sinopharm) [1,3,6,7]. From a global public health perspective, the primary and extensively studied benefit of vaccines is their efficacy in preventing disease or severe outcomes (reducing hospitalizations), ranging from 70 to 95% [3], varying based on the population and the investigated vaccine type [3].

When vaccination fails to prevent hospitalization in vaccinated individuals, an understudied scenario arises, and existing reports contradict whether being vaccinated modifies the mortality of these patients [7,8,9,10,11,12]. Additionally, how vaccines alter the clinical course and predictors of death in patients with severe/critical COVID-19 remains unknown. This is a pertinent topic that may vary not only with the type of vaccine but also in the context of the analyzed population. Mortality in hospitalized COVID-19 patients worldwide has varied from 1% to 52% [13], greatly influenced by the pandemic’s temporality, ethnic and sociocultural characteristics of each population, and the diverse therapeutic strategies applied in different regions [14].

From March 2020 to August 2022, in Mexico, the overall hospital case fatality rate was 45.1% (95% CI 44.9, 45.3), with variations depending on the pandemic period, reaching a maximum peak of 50.8% overall [6], including a peak mortality rate of 60% in hospitalized individuals aged 60 or older [6]. This places Mexico among the populations with the highest mortality in hospitalized COVID-19 patients globally. In this challenging population context, a cohort study was conducted to understand the effects of administering different types of vaccines on the clinical evolution of hospitalized patients and whether such vaccines induce changes capable of modifying the risk of death.

## 2. Materials and Methods

### 2.1. Study Subjects

An ambispective cohort study was conducted with patients admitted to the COVID-19 unit at General Hospital Number 1 of IMSS-Colima, Mexico, experiencing severe or critical COVID-19 between January 2021 and December 2022. Inclusion criteria encompassed men and nonpregnant women, aged ≥ 18, with confirmed severe or critical COVID-19 through positive SARS-CoV-2 RT-PCR or antigen test results. Patients admitted to regular hospital floors, high-flow oxygen rooms, or intensive care units (ICUs) were included, while those solely receiving care in the emergency room without admission were excluded. Subjects lacking information on vaccination status or with incomplete clinical records were eliminated. Considering the above in a cohort of 1747 hospitalized individuals, 514 were included based on predetermined inclusion and exclusion criteria. Notably, 288 of these patients (56.1%) had not received vaccination, while the remaining 226 (43.9%) had been vaccinated (Figure 1).

The study received approval from the local health research committee of IMSS-Colima, Mexico (approval number R-2020-601-041, 24 September 2020). As data were obtained from clinical records, and not directly from patients, the ethics committee waived the requirement for signed consent from each subject. Anonymity was ensured, and personal identification was concealed in the collected databases used for analysis.

### 2.2. Measures and Follow-Up

Data were extracted from patients’ clinical records, encompassing personal history, vaccination history against COVID-19, and clinical parameters throughout their hospitalization until discharge, either due to improvement or death. The following variables were universally collected: age, sex, personal history (comorbidities, Charlson comorbidity index score), COVID-19 vaccination history (brand, number of doses, time since last dose), history of previous COVID-19 infections, smoking status (considering only current smokers per the Glossary of the National Health Interview Survey of the United States of America) [15], admission disease phase (viral/pulmonary/hyperinflammatory) [16], and clinical, laboratory, and imaging data for each day of hospital stay, along with the reason for discharge (death or improvement).

Repetitively collected data during hospitalization included: blood cell counts (neutrophils, lymphocytes, platelets), inflammation markers (neutrophil/lymphocyte ratio—NLR, Ferritin, Erythrocyte Sedimentation Rate, and C-Reactive Protein Level), glucose, creatinine, liver enzymes (alkaline phosphatase—ALP, aspartate amino transferase—AST, alanine aminotransferase—ALT, and Lactate Dehydrogenase—LDH), prothrombin time expressed as an international normalized ratio (INR), development of acute kidney disease (AKI), use of mechanical ventilation or hemodialysis, administration of medications (paracetamol, anticoagulants, antibiotics, amine support, steroids, and diuretics), and the calculation of risk scores using various clinical parameters such as COVID-GRAM, National Early Warning Score 2 (NEWS-2), and Pneumonia Severity Index (PSI). The estimated glomerular filtration rate (eGFR) was determined from serum creatinine, following the CKD-EPI 2021 equation [17]. These data were repeatedly collected throughout the entire hospital stay.

### 2.3. Sample Size

The sample size for this study was determined based on the observation of mortality differences between various groups, akin to effective pharmacological interventions. The calculation was derived from previously reported mortality rates following the use or non-use of corticosteroids (41.3% and 24.8%, respectively) in hospitalized COVID-19 patients requiring high-flow oxygen [18]. To achieve the required power of 0.8, 126 patients were needed in each group (vaccinated and unvaccinated). Following the study’s completion, a post hoc statistical power analysis was conducted, revealing that being vaccinated (with two doses of BNT162b2 (Pfizer—BioNTech) or ChAdOx1-S (AstraZeneca) vaccines) reduces mortality in hospitalized COVID-19 patients, resulting in a statistical power of 93.3% for the vaccinated group and 96% for the subgroup vaccinated with BNT162b2.

### 2.4. Statistical Analysis

Data representation utilized percentages and mean ± standard deviation. The normality of data was assessed with Kolmogorov-Smirnov tests. Fisher’s exact tests or likelihood ratio chi-square tests were employed for comparing categorical data among groups. When applicable, independent Student’s t-tests or ANOVA were used to compare numerical data between two or three groups. The values of various clinical characteristics throughout the entire hospitalization period were analyzed for their predictive ability regarding patient death, utilizing the area under the Receiver Operating Characteristic (ROC) curve (AUC), confidence interval, cut-off point, sensitivity, and specificity.

Association analysis was conducted using multivariate generalized linear mixed models (GLMM, GENLINMIXED in SPSS) with a binary logistic regression link and separate random intercepts (SPRI), as previously described [19,20,21]. Data were summarized as relative risks (RRs) with 95% confidence intervals (CIs) and *p*-values, adjusted for multiple variables. GLMM is a valid strategy for estimating RRs in multivariate analysis [21,22].

Two random variables were incorporated to account for the longitudinal nature of the data: (I) day of hospital stay, and (II) month of hospital admission (pandemic time, month 1 January 2021, to month 24 December 2022). The target variable was the patient’s death during hospital stays (dichotomous; yes or no). Fixed effects included continuous variables (Charlson Index) and dichotomous variables indicating various clinical characteristics and vaccine types. Numerical clinical variables were dichotomized based on the cut-off point obtained from ROC curve calculations predicting patient death. Covariance structures were selected based on Akaike’s Information Criterion (AIC). The primary objective of the model was to obtain marginal risk by summarizing binomial regression parameters into relative risk (RR) with 95% confidence intervals (CI) and *p*-values. Univariate linear mixed effects model tests were used to compare the evolution of clinical parameters between different vaccine groups (fixed effects) during the hospitalization period (repeated observations), employing the two random variables described earlier ). Statistical power and sample size were calculated using CinCalc version 1 (https://clincalc.com/stats/Power.aspx (16 September 2023)) [21]. All other analyses were performed using SPSS Statistics version 20 software (IBM Corp., Armonk, NY, USA). A significance level of *p* < 0.05 was considered statistically significant.

## 3. Results

### 3.1. Patient Characteristics

Five hundred and fourteen hospitalized patients were enrolled in the study. Among them, 288 (56.1%) had not received any COVID-19 vaccination, while 226 (43.9%) had received at least one vaccine dose. Table 1 illustrates the vaccination schemes received. It is evident that most patients were unvaccinated, followed by those vaccinated with ChAdOx1-S (Oxford-AstraZeneca) and BNT162b2 (Pfizer—BioNTech). For the analyses of clinical evolution and prognosis, only these three patient subgroups were considered, excluding those with an incomplete vaccination regimen (such as a single dose of ChAdOx1-S or BNT162b2 or less than 14 days since their second vaccine dose) or those who received other minimally represented vaccine types. Table 2 displays the key clinical characteristics of the 462 patients included in subsequent analyses (Table 2). These patients collectively accounted for a total of 4455 days of hospital stay. Only two (0.43%) of these 462 patients reported having had a previous COVID-19 infection, so this characteristic was not included as a variable in the analyses.

### 3.2. Differences in Clinical Characteristics between Vaccinated and Unvaccinated Individuals

Table 2 compares various clinical characteristics at admission and during hospital stay between the groups of patients who were unvaccinated and those vaccinated with ChAdOx1-S or BNT162b2 (complete regimen, at least two doses). Vaccinated individuals exhibit more comorbidities (diabetes, hypertension, CKD, Charlson index, and low eGFR) than the unvaccinated. Notably, up to 90% of BNT162b2-vaccinated patients are aged over 60, whereas, in the unvaccinated or ChAdOx1-S-vaccinated groups, this age group represents 57% of patients. This aligns to the vaccination strategy in Mexico, where BNT162b2 vaccine was initially exclusively administered to healthcare personnel and those over 60, followed by the widespread administration of the ChAdOx1-S vaccine, with initial preference given to patients with comorbidities and those over 60 [23]. The body mass index (BMI) was not different between the three groups (*p* = 0.264, ANOVA test) (Table 2), so it would not be a factor that could differentiate the clinical evolution between them.

In respect to the disease stage at hospital admission, ChAdOx1-S or BNT162b2-vaccinated individuals had a higher proportion of patients in the viral (early) stage compared to the unvaccinated (viral stage at 27.7%, 60.3%, and 45.0%, respectively, ANOVA *p* < 0.001). They also have lower severity according to the NEWS-2 scale (7.7 ± 3.5, 6.5 ± 3.1, 6.6 ± 3.0, respectively, ANOVA *p* = 0.003) (Table 2). However, BNT162b2-vaccinated patients tend to have a higher proportion of patients admitted in the hyperinflammatory stage and a higher PSI score than the other groups, although, these differences were not statistically significant (see Table 2).

Regarding the treatment regimens received during hospitalization, there were no differences between the groups in the use of paracetamol, anticoagulants, antibiotics, steroids, diuretics, or amine support. However, unvaccinated individuals were more frequently required mechanical ventilation compared to those vaccinated with ChAdOx1-S or BNT162b2 (38.1%, 23.3%, 21.8%, respectively, *p* = 0.002, ANOVA test). ChAdOx1-S-vaccinated patients had a higher incidence of AKI compared to the unvaccinated (*p* = 0.001) and BNT162b2-vaccinated (*p* = 0.004) individuals (35.6%, 16.7%, 18.7%, respectively), while the latter two groups did not significantly differ (*p* = 0.545). Consequently, the need for hemodialysis was more frequent in ChAdOx1-S-vaccinated patients (ChAdOx1-S vs. unvaccinated *p* = 0.015, ChAdOx1-S vs. BNT162b2 *p* = 0.003, BNT162b2 vs. unvaccinated *p* = 0.139) (see Table 2).

During the study period (January 2021 to December 2022), various variants of COVID-19 affected the population of Mexico (research site) [24,25,26]. Considering the predominant variants, Figure 2 shows that the severity of the disease did not vary significantly in non-vaccinated patients during different periods (*p* = 0.080, ANOVA test). However, in individuals vaccinated with ChAdOx1-S (*p* < 0.001, ANOVA test) and BNT162b2 (*p* < 0.001, ANOVA test), there were significant variations between periods, with higher severity observed during periods when the Delta variant was predominant in the population.

### 3.3. Hospitalization Days and Survival

The mortality among fully vaccinated individuals was 36.2%, compared to 52.2% among the unvaccinated (*p* < 0.001). Table 3 displays the hospital stay details. The number of hospitalization days did not differ significantly among the three groups (*p* = 0.207). Overall, the highest mortality was observed in the unvaccinated group (52.6%), followed by those vaccinated with ChAdOx1-S (41.1%) and BNT162b2 (31.7%) (see Table 3). In patients aged ≥ 60 years, BNT162b2-vaccinated individuals had significantly lower mortality (33.0%) compared to those vaccinated with ChAdOx1-S (50.0%) or the unvaccinated (61.8%) (see Table 3).

BNT162b2-vaccinated individuals also experienced lower mortality than the unvaccinated when their admission disease was severe (pulmonary or hyperinflammatory phase) (43.4% vs. 59.8%, *p* = 0.010) or with a worse prognosis (PSI score > 120) (67.6% vs. 85.8%, *p* = 0.016) (see Table 3). In contrast, among patients with these characteristics, those vaccinated with ChAdOx1-S did not statistically differ from the unvaccinated (see Table 3).

In a 40-day survival follow-up (Kaplan-Meier curves), significant differences were observed among various vaccination schemes (*p* = 0.003, multigroup log-rank test). BNT162b2-vaccinated individuals exhibited higher survival rate (mean 30.3 days, 95% CI 27.3–33.2, survivors 68.3%), while the unvaccinated had lower survival rate (mean 24.3 days, 95% CI 22.5–26.1, survivors 47.4%), with ChAdOx1-S-vaccinated individuals falling in between (mean 27.2 days, 95% CI 23.6–30.8, survivors 58.9%) (see Figure 3A). Stratifying patients based on different characteristics revealed differences among groups for those hospitalized in advanced stages of the disease (pulmonary and hyperinflammatory) (*p* = 0.036, multigroup log-rank test). BNT162b2-vaccinated individuals showed higher survival rate (mean 26.6 days, 95% CI 22.3–31.1, survivors 56.6%), while ChAdOx1-S-vaccinated individuals had lower survival rate (mean 17.4 days, 95% CI 11.9–22.8, survivors 27.6%), and the unvaccinated drop in between (median 22.0 days, 95% CI 19.9–24.1, survivors 40.2%) (see Figure 3B).

On the other hand, among patients hospitalized at an early stage (viral phase), no significant differences in survival were found based on their vaccination scheme (*p* = 0.124, multigroup log-rank test). However, BNT162b2 or ChAdOx1-S-vaccinated patients had higher survival rate (not statistically significant) than the unvaccinated (80.9%, 79.5%, and 65.8%, respectively) (see Figure 3C). According to the PSI score, in patients hospitalized with values below 120 (indicating a better prognosis than higher scores, Figure 3D), BNT162b2-vaccinated individuals had higher survival rate (mean 36.7 days, 95% CI 34.3–39.1, survivors 88.9%), while the unvaccinated had lower survival (mean 30.1 days, 95% CI 27.9–32.3, survivors 67.8%), with ChAdOx1-S-vaccinated individuals falling in between (median 33.6 days, 95% CI 30.0–37.2, survivors 78.3%) (*p* = 0.001, multigroup log-rank test). Among patients with a very poor prognosis at admission (PSI > 120), no significant differences in survival were found based on their vaccination scheme (*p* = 0.395, multigroup log-rank test). However, BNT162b2 or ChAdOx1-S-vaccinated patients had better survival rates than the unvaccinated (32.4%, 24.0%, and 14.2%, respectively) (see Figure 3E).

Among patients aged 60 years or older (Figure 3F), BNT162b2-vaccinated individuals had the highest survival (mean 29.9 days, 95% CI 26.8–33.2, survivors 67.0%), while the unvaccinated had the lowest survival (mean 21.7 days, 95% CI 19.3–24.0, survivors 38.2%), with ChAdOx1-S-vaccinated individuals falling in between (mean 25.0 days, 95% CI 20.3–29.7, survivors 50.0%) (*p* < 0.001, multigroup log-rank test). Among patients younger than 60 years (Figure 3F), no significant differences were found based on their vaccination scheme (*p* = 0.386, log-rank test), although once again, BNT162b2 or ChAdOx1-S-vaccinated patients had better survival rates than the unvaccinated (80.0%, 71.0%, and 59.7%, respectively).

### 3.4. Risk Factors for Death in Hospitalized Patients with COVID-19, including Vaccination Schedule

Table 4 presents a multivariate generalized linear mixed model with binary logistic regression analysis that identifies factors associated with death in patients hospitalized with COVID-19, including the impact of their vaccination schedule. This analysis included factors that were found to be different among groups according to Table 2. The statistical model also incorporated the period of the pandemic during which the patient was hospitalized as a random effect (see Materials and Methods section). Regarding pre-existing conditions or conditions at hospital admission, smoking (RR 4.5, 95% CI 2.0–10.1, *p* < 0.001), and advanced disease phase (pulmonary/hyperinflammatory) significantly increased the risk of death (RR 4.01, 95% CI 2.4–6.5, *p* < 0.001). During hospitalization, elevated indices of severe disease, including NEWS-2 score ≥ 12 (RR 3.56, 95% CI 1.6–7.7, *p* = 0.001), PSI score > 120 (RR 8.14, 95% CI 5.0–13.1, *p* < 0.001), the need for mechanical ventilation (RR 3.74, 95% CI 1.8–7.3, *p* < 0.001), and neutrophils ≥ 8 × 10e^3^/uL (RR 3.76, 95% CI 2.3–6.0, *p* < 0.001), were identified as risk factors (see Table 4). Additionally, having lymphocytes ≥680/uL (RR 0.53, 95% CI 0.3–0.8, *p* = 0.007) or ALT > 45 UL (RR 0.45, 95% CI 0.2–0.8, *p* = 0.005) were identified as protective factors. BNT162b2-vaccinated individuals had a 2.4-times reduced risk of death (RR 0.41, 95% CI 0.2–0.8, *p* = 0.008), whereas ChAdOx1-S-vaccinated individuals showed no significant difference in the risk of death compared to the unvaccinated (RR 1.04, 95% CI 0.5–2.3, *p* = 0.915) (see Table 4). The receipt of a booster dose was also not associated with the risk of death (*p* = 0.247) (see Table 4). Combining the patients vaccinated with BNT162b2 and whereas ChAdOx1-S into a single group, it is observed that patients in this group reduce their risk of death by almost half, with a RR value of 0.54 (95% CI 0.30–0.97), being a congruent value as it is located between the RR values of BNT162b2 (0.41) and whereas ChAdOx1-S (1.04). It is important to highlight those only patients with a complete vaccination schedule (at least two doses) were considered, and all booster vaccinations were administered as ChAdOx1-S, irrespective of the initial vaccination scheme.

### 3.5. Clinical Differences between Those Vaccinated with BNT162b2 and ChAdOx1-S

In Table 5, a multivariate analysis is shown to determine the probability of being vaccinated with BNT162b2 according to the presence of various characteristics, compared to patients vaccinated with ChAdOx1-S [21,22]. It is observed that being > 60 years old, being admitted in an advanced phase of the disease (pulmonary/hyperinflammatory) and having a PSI score > 120 were factors associated with being vaccinated with BNT162b2, in comparison to patients vaccinated with ChAdOx1-S. Conversely, having ferritin levels >810 ng/mL, receiving an additional vaccine dose, or experiencing death were variables less likely to be present in those vaccinated with BNT162b2 (see Table 5). This makes it evident that patients vaccinated with BNT162b2 had 3.7-times lower odds of death than patients vaccinated with ChAdOx1-S, despite being older and having a more advanced and severe disease at admission. Among the hematological or inflammatory parameters that could explain these differences between different vaccination schemes, the protection against elevated ferritin levels in patients with BNT162b2 stands out.

In Table 6, a multivariate analysis is shown to determine the probability of being vaccinated with Pfizer according to the presence of various characteristics, compared to non-vaccinated patients. It shows that being over 60 years old, male, having a higher comorbidity index, and having higher levels of neutrophils and lymphocytes were factors associated with being vaccinated with BNT162b2 compared to non-vaccinated patients. On the contrary, not smoking, experiencing less acute kidney injury during hospitalization, or dying were variables less likely to be present in BNT162b2-vaccinated individuals (see Table 6). This makes it clear that BNT162b2-vaccinated patients died 2.6-times less than non-vaccinated patients, despite being older and having more comorbidities. Having less lymphopenia and the presence of acute kidney injury appear to be the most relevant factors that could explain the lower mortality in BNT162b2-vaccinated patients compared to non-vaccinated individuals.

### 3.6. Evolution of Some Clinical Parameters during the First 6 Days

The evolution of some risk factors for mortality (see Table 4) during the first 6 days of hospitalization was analyzed. In Figure 4A, the NEWS-2 score is higher in non-vaccinated patients at admission compared to vaccinated patients (*p* = 0.026; and *p* = 0.018; compared to ChAdOx1-S and BNT162b2, respectively). However, on day 4, patients vaccinated with ChAdOx1-S begin to increase this score, and those vaccinated with BNT162b2 remain with the lowest levels throughout all the days analyzed. On day 6, the NEWS-2 score was higher in non-vaccinated patients, followed by those vaccinated with ChAdOx1-S and BNT162b2 (8.4 ± 4.1, 7.3 ± 3.6, 5.8 ± 3.9, respectively). This is consistent with the proportion of patients who ultimately died in each patient group. The PSI (Figure 4B) was initially higher in patients vaccinated with BNT162b2, but as the days progressed, non-vaccinated patients and those vaccinated with ChAdOx1-S began to increase their PSI until, by day 6, there were no longer differences between the groups (118.7 ± 46.1, 122.6 ± 39.3, 124.1 ± 41.9, respectively). The absolute values of lymphocytes (Figure 4C) were, on average, always lower in non-vaccinated patients compared to vaccinated ones, with no differences observed between the two vaccine types. In terms of the PLR levels, non-vaccinated patients clearly had the highest values (Figure 4D). Finally, the serum level of ferritin (Figure 4E) in patients vaccinated with BNT162b2 was on average clearly lower than the value of non-vaccinated or ChAdOx1-S-vaccinated patients. Repeated measurement data for these variables were analyzed using multivariate linear mixed-effects models, confirming a significant difference in the BNT162b2 group compared to the other two groups in PSI index and ferritin levels, while the parameters of NEWS-2 score, lymphocytes, and PLR behaved similarly in both vaccinated groups, with significant differences compared to the values of non-vaccinated patients (see Table 7).

### 3.7. Predictors of Death Based on Vaccination Status

Table 8 and Appendix A present various clinical useful factors for predicting mortality in different patient groups based on their vaccination status. The area under the ROC curve (AUC) was calculated to determine the optimal cut-off point for each variable in predicting death within each patient group. The Pneumonia Severity Index (PSI) emerged as a relevant predictor of death toll across all groups, with significantly lower AUC values in BNT162b2-vaccinated patients compared to those vaccinated with ChAdOx1-S (*p* = 0.043) or unvaccinated individuals (*p* < 0.001). It is noteworthy that the cut-off points for predicting death were higher in BNT162b2 or ChAdOx1-S-vaccinated individuals compared to those the unvaccinated (124.5, 123.5, and 118.0, respectively), demonstrating good sensitivity (close to 0.80 in all groups) but low specificity. This indicates that vaccinated patients need to reach a higher severity to predict their mortality. Regarding oxygen levels, the AUCs were approximately 0.35 in all groups, showing no differences among them (*p* > 0.05 for all intergroup comparisons). However, BNT162b2 or ChAdOx1-S-vaccinated patients had higher cut-off points than the unvaccinated individuals (82.5%, 84.5%, and 74.0%, respectively) to predict patient survival or mortality with high sensitivity and specificity in all groups, although, it was higher in the unvaccinated (see Table 7). This aligns to arterial pH analysis, where AUC values were significant in predicting death in all groups (AUC 0.25, 0.29, and 0.37; in the unvaccinated, ChAdOx1-S, or BNT162b2, respectively). However, the cut-off point was more acidic in the unvaccinated (pH 7.2305) compared to ChAdOx1-S or BNT162b2-vaccinated individuals (pH 7.3100, and 7.3005, respectively) (see Appendix A). Arterial pH was significantly lower on average in unvaccinated or ChAdOx1-S-vaccinated patients compared to BNT162b2-vaccinated patients (7.339 ± 0.16, 7.333 ± 0.15, and 7.360 ± 0.14, respectively, *p* < 0.05 for both comparisons, see Table 8). This suggests that lung damage might be more relevant in predicting death toll in unvaccinated patients than in vaccinated ones.

Regarding immune system cells, as shown in Table 7, vaccinated patients had significantly higher cell counts than the unvaccinated individuals. Simultaneously, it was observed that a deeper degree of lymphopenia was necessary to predict death in unvaccinated patients (cutoff: 535/µL) compared to ChAdOx1-S or BNT162b2-vaccinated groups (cutoff: 831/µL and 893/µL, respectively). On the other hand, for neutrophils to be a predicting factor for death, vaccinated patients needed to have neutrophilia (neutrophil count > 7700/microliter), compared to a normal value in unvaccinated individuals (cutoff in ChAdOx1-S, BNT162b2, and unvaccinated: 9711, 9430, and 7311, respectively) (see Table 8). It has been previously reported that when neutrophilia is present in a hospitalized COVID-19 patient, an increased inflammatory status and cytokine storm occurs, and potential bacterial infections should be considered [27,28]. An average neutrophil count of over 9100 cells/µL was found in patients with confirmed bacterial infection hospitalized for COVID-19. If we consider this value as a possible indicator of bacterial infection, it is found that days prior to hospital discharge, this factor was present in 49.2%, 64.6%, and 77.8% of deceased patients in unvaccinated, ChAdOx1-S-vaccinated, and BNT162b2-vaccinated individuals, respectively (*p* < 0.001, Fisher’s exact test of 2 × 3 contingency table). This was significantly lower in deceased unvaccinated individuals compared to ChAdOx1-S-vaccinated (*p* = 0.036) and BNT162b2-vaccinated individuals (*p* < 0.001), although, the two vaccinated groups were not statistically different (*p* = 0.105). With this, clinical differences between vaccinated and unvaccinated individuals are evident, indicating that neutrophilia seems to be a relevant factor in predicting death in vaccinated patients, suggesting that the presence of bacterial co-infection could be a more critical factor for mortality in vaccinated individuals.

However, it is interesting to note differences between the two vaccinated groups, considering that individuals vaccinated with BNT162b2 have over 3.7-times less likelihood of dying compared to those vaccinated with ChAdOx1-S (see Table 5). In liver function tests, the enzyme LDH emerged as the most relevant parameter for predicting death toll in all groups. A significant difference in the AUC was observed between the ChAdOx1-S group and the BNT162b2 group (AUC 0.873 vs. 0.758, *p* = 0.002) and between the ChAdOx1-S group and the unvaccinated group (ChAdOx1-S vs. unvaccinated, AUC 0.873 vs. AUC 0.744, *p* < 0.001) (Table 8). However, it is important to mention that unvaccinated individuals have a higher cut-off point than those vaccinated with ChAdOx1-S or BNT162b2 (367, 333, and 325 IU/L, respectively) (Table 8). LDH only demonstrates good sensitivity (0.78) with little specificity for predicting death in those vaccinated with ChAdOx1-S, while in those vaccinated with BNT162b2 or unvaccinated, its sensitivity is much lower (0.61 and 0.68, respectively) (Table 8). On the other hand, the enzymes ALP, ALT, and AST were not significant predicting factors in the ChAdOx1-S-vaccinated group, but they were in those vaccinated with BNT162b2 and the unvaccinated group. However, this does not imply that these enzymes are not relevant for ChAdOx1-S-vaccinated patients, as this group had, on average, higher serum levels of ALP, ALT, and AST (see Table 9). This could indicate that the values of these enzymes were elevated in both those who died and those who survived, making it an irrelevant parameter for distinguishing between these outcomes in ChAdOx1-S-vaccinated patients.

Regarding renal function, eGFR upon hospital admission showed no differences between patients vaccinated with ChAdOx1-S or BNT162b2, but these groups had higher values compared to the unvaccinated individuals (see Table 2). The eGFR value was useful for predicting death in all groups, with significantly different AUCs between the unvaccinated and ChAdOx1-S groups (0.318 vs. 0.425, *p* = 0.001) and between the unvaccinated and BNT162b2 groups (0.316 vs. 0.431, *p* < 0.001), but no difference in AUCs between the ChAdOx1-S and BNT162b2 groups (*p* = 0.865) (Table 8). However, the cutoff values were observed to be very different, at 75.2, 36.3, and 60.0 mL/min/1.73m^2^ in the unvaccinated, ChAdOx1-S-vaccinated, and BNT162b2-vaccinated groups, respectively (see Table 8). This may suggest that severe loss of renal function as a predictor of death only occurred in the ChAdOx1-S-vaccinated group (eGFR < 36), while the loss of renal function for predicting death was mild to moderate in the BNT162b2 and unvaccinated groups, indicating that other factors were likely more relevant in these latter two groups.

In markers of inflammation, there was a difference in AUCs between ChAdOx1-S and BNT162b2 for PCR (*p* = 0.002), ferritin (*p* = 0.032), and VCP (<0.001), with these three factors being significant predictors of death in BNT162b2-vaccinated patients but not in those vaccinated with ChAdOx1-S (see Table 8 and Appendix A). These three factors were also relevant for predicting death in unvaccinated patients. On the other hand, D-dimer was a useful and similar parameter for predicting death significantly in all groups (see Table 8). With all the above, it can be observed that there are some different factors between patients vaccinated with BNT162b2 or ChAdOx1-S. For those vaccinated with BNT162b2, a combination of markers of liver damage (AST, ALT, ALP, and LDH) and certain inflammation markers (ferritin, PCR, and VCP) were useful for predicting patient death, but not in those vaccinated with ChAdOx1-S. This does not mean that all these markers are low in ChAdOx1-S-vaccinated patients; on the contrary, ALP, ferritin, and D-dimer are significantly elevated in this group compared to those vaccinated with BNT162b2, with the rest of the liver enzymes and CRP following the same trend (see Table 9). On the other hand, the average values of ferritin, CRP, D-dimer and LDH do not differ between patients vaccinated with ChAdOx1-S and those unvaccinated. Additionally, the average values of ALT and ALP were significantly higher in patients with ChAdOx1-S than in those unvaccinated (see Table 9).

## 4. Discussion

In the context of a population with one of the highest mortality rates worldwide for COVID-19 hospitalized patients, there were differences in the proportion of deaths based on the vaccination scheme. Those who died in higher proportion were the unvaccinated (52.6%), followed by those vaccinated with ChAdOx1-S (41.1%) and those vaccinated with BNT162b2 (31.7%) (vaccination schemes with at least two doses and at least 14 days elapsed since their second dose). The lower mortality of those vaccinated with BNT162b2 occurred even though these patients were older and had higher severity (higher PSI) at the time of hospital admission. Those vaccinated with ChAdOx1-S or BNT162b2 had nearly half the risk of dying from the disease compared to the unvaccinated (adjusted RR 0.54, 95% CI 0.30–0.97, *p* = 0.041). However, when stratifying by vaccine type, those vaccinated with BNT162b2 had 2.4-times less risk of death compared to the unvaccinated (adjusted RR 0.41, CI95% 0.2–0.8, *p* = 0.008), while those vaccinated with ChAdOx1-S did not differ from the unvaccinated in terms of their risk of death (adjusted RR 1.04, CI95% 0.5–2.3, *p* = 0.915). This suggests differences in the clinical outcomes of patients based on the administered vaccine.

The reduction in the risk of death by vaccination in hospitalized COVID-19 patients aligns with previous reports from Croatia [8] and Argentina-Spain [7]. However, other studies have not shown differences in mortality between vaccinated and unvaccinated patients in populations from the United States [9], Israel [10], or France [11]. Even a study conducted in Poland suggests that hospitalized patients may have a higher mortality after receiving one or two vaccine doses [12]. Huespe IA et al. (2023) mention that these contradictory results may be due to confounding variables, especially differences in the proportion and severity of comorbidities, generally higher in vaccinated groups at the beginning of the pandemic [7]. Differences among subgroups of populations can also lead to variations in the immune response to vaccines, which is progressively lower with older age and a higher prevalence of comorbidities [29].

Among studies that have found that vaccination reduces mortality in hospitalized patients, one points out that those vaccinated with ChAdOx1-S (and other vaccines with adenoviral vectors) have lower mortality than those vaccinated with BNT162b2 [8], while another observed that those vaccinated with BNT162b2 had a greater reduction in mortality (OR 0.37; 95% CI: 0.23–0.59), followed by those vaccinated with ChAdOx1-S (OR 0.42; 95% CI: 0.20–0.86) [7]. In the first study, Busic et al. (2022) discuss that those vaccinated with BNT162b2 could have higher mortality than those vaccinated with ChAdOx1-S because mRNA vaccines (such as Pfizer—BioNTech) were the first vaccines to be administered, having preference for the older population and among selected patients with unfavorable prognostic characteristics [7,8]. In the present report from Mexico, something similar happened in the vaccination strategy, with older patients and those with a higher index of comorbidities initially vaccinated and using BNT162b2. Therefore, patients vaccinated with BNT162b2 had more adverse characteristics and higher severity at the time of hospital admission, especially during periods when the Delta variant was one of the predominant strains in the population (see Figure 2). However, even with a worse prognosis, those vaccinated with BNT162b2 had lower mortality in absolute numbers. In the analysis adjusting for variables such as age, sex, morbidity, severity, as well as, the pandemic period to consider variations in different waves (as a random effect), it was found that those vaccinated with BNT162b2 reduce the risk of death by 2.4-times compared to the unvaccinated, while those vaccinated with ChAdOx1-S did not differ from the unvaccinated ones in adjusted risk of death (see Table 4). However, as mentioned above, both those vaccinated with ChAdOx1-S and BNT162b2 reduce the overall percentage of death compared to those not vaccinated (41.1%, 31.7%, and 52.6%, respectively).

In hospitalized patients, it was observed that receiving either the ChAdOx1-S or BNT162b2 vaccine contributes to higher lymphocyte counts and lower Platelet-to-Lymphocyte Ratios (PLR) compared to the unvaccinated group (Figure 4, Table 2, Table 7 and Table 9). Vaccination prevented severe hypoxemia from being the most sensitive and specific factor for predicting death, as observed in the unvaccinated group. This aligns to the improved survival rates found in both vaccinated groups. It is noteworthy that in the vaccinated individuals, neutrophilia serves as a predictive factor for death, suggesting that a concomitant bacterial infection might be a relevant factor leading to mortality in the vaccinated group. However, individuals vaccinated with BNT162b2 have a lower risk of death than those vaccinated with ChAdOx1-S (Table 5). This may be attributed to the additional observation that, on average, individuals vaccinated with BNT162b2 exhibit less impairment in markers of hepatic damage, lower levels of certain inflammatory markers (such as serum ferritin), and lower levels of D-dimer compared to those vaccinated with ChAdOx1-S or the unvaccinated group.

The vaccines appear to induce a more robust immune response against severe/critical COVID-19 infection, leading to increased survival. The results suggest that more severe damage to other organs or systems, in addition to the lungs, would be necessary to predict death in vaccinated patients (Table 8, Table 9 and Appendix A). However, the ChAdOx1-S vaccine has a less beneficial effect than BNT162b2 in hospitalized patients. One possible explanation for this could be a lower stimulation of the immune system against COVID-19, resulting in lower survival rates in severe patients. It has been previously demonstrated that the efficacy for preventing disease or severe disease is higher for BNT162b2 compared to ChAdOx1-S (95–87.5% vs. 70%, respectively) [3], which aligns with a lower neutralization of SARS-CoV-2 by serum from individuals vaccinated with ChAdOx1-S compared to serum from BNT162b2-vaccinated individuals [3]. Another potential cause could be that using an adenoviral vector as a vehicle to stimulate immunity may limit some vaccine benefits. Exposure to adenovirus in preclinical models has led to long-term effects, including adverse metabolic, morphological (hepatic), and functional changes, with significantly high levels of serum inflammation markers [30,31]. It has been postulated that these chronic adenoviral effects, including prolonged inflammatory responses [32,33], could occur with their application in gene therapy [30] or after their use in COVID-19 vaccines [34]. However, these are unverified hypotheses that need to be investigated in future research, as the emergency authorization of COVID-19 vaccines (adenoviral or mRNA) may not have fully addressed certain safety aspects [32,35].

A strength of the present study was the adjustment for comorbidities, severity, and other relevant factors that can affect the death prognosis of patients, as well as the analysis with repeated measurements of various clinical factors during hospitalization. Additionally, the temporality of the pandemic (month of hospital admission) was considered in the statistical model used. Another relevant aspect is that the effect of vaccination was analyzed from different points of view (mortality rate, survival curves, association analysis, ROC curves, evaluation of hematological and biochemical parameters), so conclusions can be reached based multiple aspects and not with the result of a single analysis. Undoubtedly, evaluating the effect of vaccination in a population of hospitalized patients with high mortality helps assess the potential impact of vaccines under very adverse prognostic conditions, which is challenging to evaluate in other populations or periods of the pandemic.

Considering the results obtained in this study, it is imperative to highlight that these findings reinforce the efficacy of COVID-19 vaccines in reducing hospital fatalities, in addition to the previously reported reduction in severity and hospitalizations [3,4]. However, addressing these findings with caution is fundamental to avoid misinterpretations that could be misused for anti-vaccine purposes. It is crucial to emphasize that this study not only highlights a significant decrease in the risk of mortality for vaccinated individuals, especially in populations with increased health risks, but also underscores notable differences among the various types of vaccines used. The findings highlight the crucial and beneficial role of vaccination, adding to what was previously described to reduce symptoms and the risks of hospital admission and death [36,37]. These collective results underscore the current importance of vaccination efforts, considering variations in populations and the types of vaccines used.

The present study also had limitations. The number of patients did not allow for evaluations and comparisons between patients with a single vaccine dose. Similarly, the mortality in patients vaccinated with ChAdOx1-S was intermediate between the unvaccinated and those vaccinated with BNT162b2, so in various analyses, statistically significant differences between ChAdOx1-S and the other two groups were not observed. Future analyses with a larger number of individuals are desirable to better observe the differences between these groups. Additional facets of the investigation pertinent to the studied population, such as the types of vaccines administered and who was given priority for vaccination at the beginning of the pandemic in Mexico, did not allow for the evaluation of other vaccine brands and types as well as their effects on various population strata with fewer comorbidities. Likewise, only 8.9% of the hospitalized patients included had a booster vaccine, making a detailed analysis of this subgroup unfeasible. It is important to note that exploring this aspect was limited in the current study. Future research efforts should consider a more in-depth examination of this subgroup, especially in the context of different COVID variants.

## 5. Conclusions

In summary, in the context of a population of hospitalized patients with COVID-19 with a very adverse prognosis and very high mortality, patients vaccinated with BNT162b2 had a lower risk of death compared to those vaccinated with ChAdOx1-S and the unvaccinated. Patients vaccinated with ChAdOx1-S had an intermediate mortality rate between BNT162b2-vaccinated patients (who had the lowest mortality) and the unvaccinated group (with the highest mortality). The protective effect observed in vaccinated patients can be explained by higher lymphocyte counts and lower levels of some inflammation markers such as PLR. However, patients vaccinated with BNT162b2 additionally have lower values of other inflammation markers, such as ferritin and D-dimer. Unlike unvaccinated patients, it can be inferred that, in general, vaccinated patients require more extensive damage to other organs, in addition to lung damage, or other complications (such as concomitant bacterial infection) to predict death toll. More studies on the effects of COVID-19 vaccination are necessary.

## Figures and Tables

**Figure 1 vaccines-12-00072-f001:**
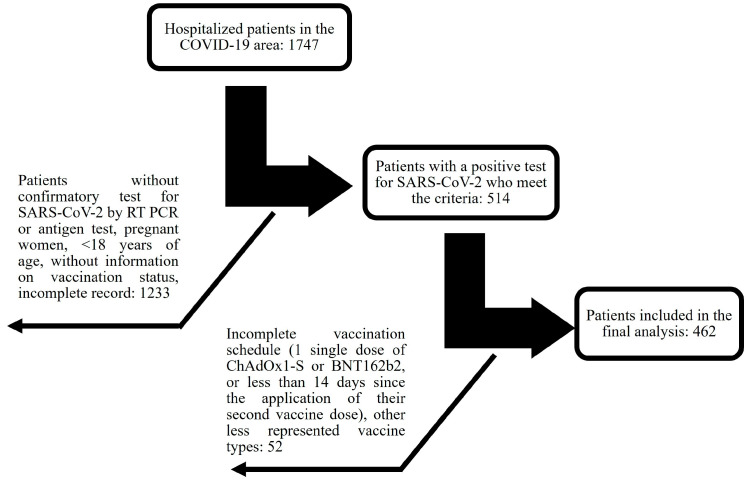
Flowchart depicting the selection process from a total of 1747 hospitalized individuals according to inclusion and exclusion criteria.

**Figure 2 vaccines-12-00072-f002:**
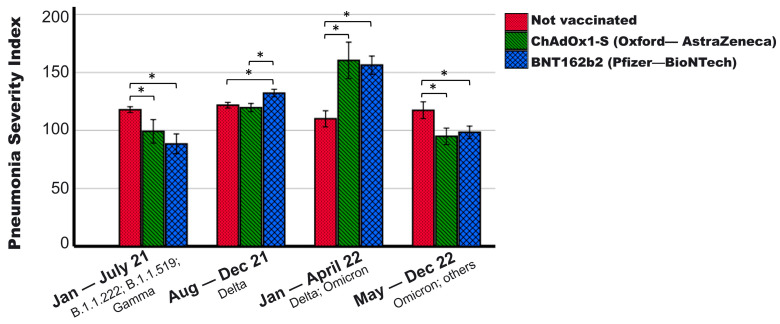
Pneumonia Severity Index of patients with different vaccination schemes according to the pandemic period. It is observed that vaccinated patients show variations in severity depending on the pandemic period, while the severity of the non-vaccinated remained constant. In the period when the Delta variant was predominant, vaccinated patients experienced higher severity than the non-vaccinated, exhibiting the opposite trend in other periods. Periods were established based on the predominant variants in the general population of Mexico ([24,25,26]; January–July 2021: B.1.1.222; B.1.1.519; and Gamma. August–December 2021: Delta. January–April 2022: Delta, Omicron BA.1; Omicron BA.1.1, Omicron BA.1.15. May–December 2022: Omicron, others. * *p* < 0.01. Mean + 95% CI is shown.

**Figure 3 vaccines-12-00072-f003:**
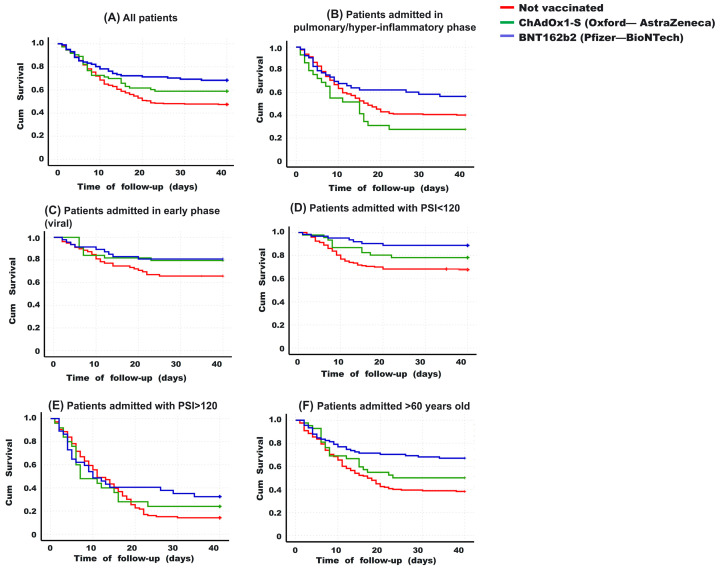
Kaplan-Meier curves for 40 days based on vaccination scheme. (**A**) Overall, BNT162b2-vaccinated patients show better survival than the other groups (multiple comparisons *p* = 0.003. Post-hoc analysis: *p* = BNT162b2 vs. unvaccinated *p* = 0.001. ChAdOx1-S vs. unvaccinated *p* = 0.130. BNT162b2 vs. ChAdOx1-S *p* = 0.223). (**B**) Among those hospitalized in advanced stages (pulmonary and hyperinflammatory phase), a similar trend is observed (multiple comparisons *p* = 0.036. Post-hoc analysis: BNT162b2 vs. unvaccinated *p* = 0.076. ChAdOx1-S vs. unvaccinated *p* = 0.120. BNT162b2 vs. ChAdOx1-S *p* = 0.014. (**C**) In patients hospitalized in an early stage (viral phase), no significant differences were found between the groups (multiple comparisons *p* = 0.124). (**D**) In patients hospitalized with PSI values < 120, BNT162b2-vaccinated individuals had higher survival (multiple comparisons *p* = 0.001. Post-hoc analysis: BNT162b2 vs. unvaccinated *p* = 0.002. ChAdOx1-S vs. unvaccinated *p* = 0.161. BNT162b2 vs. ChAdOx1-S *p* = 0.137). (**E**) When having a PSI > 120 at hospital admission, the groups did not differ (multiple comparisons *p* = 0.395). (**F**) Among patients aged 60 years or older, BNT162b2-vaccinated individuals had higher survival than the other groups (multiple comparisons *p* < 0.001. Post-hoc analysis: BNT162b2 vs. unvaccinated *p* = 0.002. ChAdOx1-S vs. unvaccinated *p* = 0.161. BNT162b2 vs. ChAdOx1-S *p* = 0.137), while in those under 60 years, no differences were found between the groups (multiple comparisons *p* = 0.386). The log-rank test was applied to compare curves in multiple comparisons or post-hoc analysis.

**Figure 4 vaccines-12-00072-f004:**
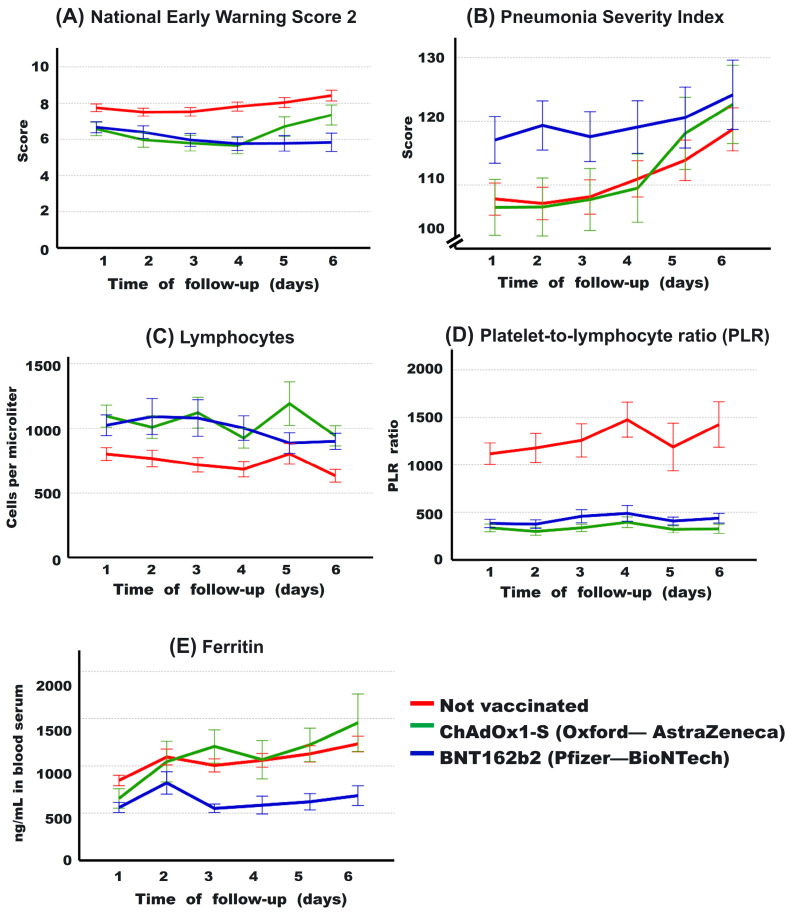
Evolution of some clinical parameters during the first 6 days of hospitalization for patients based on the type of vaccine received. It is observed that the National Early Warning Score 2 (NEWS-2) is lower in the vaccinated groups compared to the non-vaccinated group (**A**). In contrast, regarding the Pneumonia Severity Index (PSI), BNT162b2-vaccinated individuals show higher severity (**B**). However, it is noticeable that both ChAdOx1-S-vaccinated individuals (in (**A**,**B**)) and the non-vaccinated group experience an increase over the course of days (**B**). The lymphocyte count (**C**) is higher in vaccinated individuals, with a markedly lower platelet-to-lymphocyte ratio (**D**) in these groups compared to the non-vaccinated group. Ferritin levels (**E**) are lower in BNT162b2-vaccinated individuals compared to those vaccinated with ChAdOx1-S and the non-vaccinated group. Average values and 95% confidence intervals are presented.

**Table 1 vaccines-12-00072-t001:** Data on the COVID vaccination scheme in hospitalized patients.

Clinical Characteristic		Vaccinated		
	All	No	Yes	*p*
	n = 514	n = 288	n = 226	
Medical personal history				
Age (years)	63.3 ± 16.1	62.2 ± 16.3	64.6 ± 15.7	0.084 *
≥60 years	63.1%	57.1%	70.8%	0.001 **
Male (%)	61.9%	61.5%	62.4%	0.451 **
Data on the COVID vaccination scheme				
Vaccinated		56.0%	44.0%	
ChAdOx1-S ^a^			17.9%	
BNT162b2 ^b^			21.7%	
Ad5-nCoV			2.7%	
CorovaVac ^c^			1.7%	
Complete primary series			38.2%	
Booster vaccinations ^d^			8.9%	
Days since last application			132.9 ± 97.0	

Percentages or averages and standard deviation are shown. * Independent Student’s *t*-test analysis. ** Fisher’s exact tests. ^a^ Oxford—AstraZeneca. ^b^ Pfizer—BioNTech. ^c^ Sinovac. ^d^ Only ChAdOx1-S (Oxford–AstraZeneca) was used.

**Table 2 vaccines-12-00072-t002:** Main clinical characteristics of the participating subjects at the time of enrollment and prescribed drugs.

Clinical		Vaccinated				*p* *	
Characteristic	No	ChAdOx1-S	BNT162b2	Inter-	No vs.	No vs.	ChAdOx1-S
	n = 288 (100%)	n = 73 (100%)	n = 101 (100%)	Group	ChAdOx1-S	BNT162b2	vs. BNT162b2
≥60 years	57.1%	57.5%	90.1%	<0.001	0.527	<0.001	<0.001
Male (%)	61.5%	61.6%	65.3%	0.777			
Diabetes	38.0%	54.8%	50.5%	0.009 *	0.007	0.019	0.342
HBP	34.3%	56.2%	54.5%	<0.001 *	0.001	<0.001	0.473
BMI	30.4 ± 7.2	29.3 ± 4.2	30.6 ± 7.5	0.264			
Smoking	9.6%	1.4%	6.1%	0.025 *	0.013	0.197	0.135
CKD	16.3%	40.3%	29.7%	<0.001	<0.001	0.003	0.100
Charlson Index	3.2 ± 2.1	4.2 ± 2.1	4.7 ± 1.7	<0.001 *	0.001	<0.001	0.106
Clinical data at hospital admission							
PSI	107 ± 42	106 ± 37	117 ± 36	0.110			
Disease phase				<0.001	<0.001	<0.001	0.058
Viral	27.7%	60.3%	45.0%				
Pulmonary	57.1%	28.8%	30.0%				
Inflammation	15.2%	11.0%	25.0%				
C-GRAM	135 ± 36	131 ± 41	134 ± 32	0.692			
NEWS-2	7.7 ± 3.5	6.5 ± 3.1	6.6 ± 3.0	0.003	0.011	0.007	0.843
Neutrophils	7309 ± 5972	10,604 ± 7244	9606 ± 6101	<0.001	0.001	0.001	0.336
Lymphocytes	801 ± 793	1094 ± 714	1023 ± 789	0.004	0.005	0.019	0.554
PlateletsX1000	261 ± 122	267 ± 147	244 ± 103	0.390			
NLR	13.0 ± 12.5	15.3 ± 21.6	13.6 ± 12.9	0.487			
D-Dimer	2999 ± 2184	3142 ± 5579	2180 ± 3583	0.914			
ESR	29.8 ± 11.6	34.5 ± 16.8	23.8 ± 11.7	0.050	0.226	0.047	0. 045
CRP	17.1 ± 18.6	15.5 ± 11.2	16.6 ± 28.8	0.971 *			
Ferritin	845.8 ± 638.5	655.2 ± 586.2	559.7 ± 336.1	0.017	0.122	0.009	0.394
Creatinine	2.2 ± 3.7	3.8 ± 4.4	2.4 ± 3.2	0.006	0.003	0.760	0.014
eGFR	73.9 ± 48.7	51.9 ± 58.3	54.1 ± 36.7	<0.001	0.003	0.002	0.999
AST	50.1 ± 51.2	125.6 ± 397.6	67.5 ± 193.2	0.046	0.007	0.328	0.329
ALT	41.2 ± 38.8	83.2 ± 216.2	42.4 + 122.8	0.035	0.008	0.900	0.195
ALP	99.3 ± 65.6	139.8 ± 170.5	103.1 ± 70.8	0.068			
LDH	384.2 ± 209.5	502.4 ± 954.6	345.8 ± 288.0	0.078			
Glucose	191.5 ± 132.3	179.5 ± 129.2	214.9 ± 169.5	0.232			
INR	1.14 + 0.25	1.13 + 0.31	1.19 + 0.59	0.690			
Main treatments during hospital stay							
AKI	16.7%	35.6%	18.7%	0.002	0.001	0.545	0.004
Paracetamol	11.4%	11.0%	4.0%	0.054			
Anticoagulants	89.3%	86.3%	94.1%	0.193			
Antibiotics	48.7%	42.3%	49.0%	0.746			
Amine support	9.7%	6.8%	8.0%	0.689			
Steroids	90.6%	93.2%	93.1%	0.642			
Diuretics	14.6%	29.5%	22.1%	0.150			
Mech. Vent.	38.1%	23.3%	21.8%	0.002	0.012	0.002	0.478
Hemodialysis	9.0%	19.2%	5.0%	0.010	0.015	0.139	0.003

Percentages or averages and standard deviation are shown. * To compare three groups (intergroup analysis), ANOVA or likelihood ratio chi-square test was performed, to compare two groups, independent Student’s *t*-test or Fisher’s exact test was used, as appropriate for numerical or qualitative data, respectively. BMI: Body mass index. HBP: Systemic arterial hypertension. Smoking: current smoker. CKD: Chronic kidney disease. PSI: Pneumonia Severity Index. C-GRAM: Critical Illness Risk Score (COVID-GRAM). NEWS-2: National Early Warning Score 2. Neutrophils, lymphocytes, and plateletsX1000, expressed in cells per microliter of blood. NLR: neutrophil/lymphocyte ratio: D-Dimer expressed in ng/mL. ESR: Erythrocyte Sedimentation Rate, mm/hour. CRP: C-Reactive Protein mg/dL. Ferritin expressed in ng/mL. Creatinine expressed in mg/dL. eGFR: estimated glomerular filtration rate, expressed in mL/min/1.73 m^2^. ALP: alkaline phosphatase, expressed in international units per liter (IU/L). AST: aspartate amino transferase, expressed in IU/L. ALT: alanine aminotransferase, expressed in IU/L. LDH: Lactate Dehydrogenase expressed in IU/L. Glucose expressed in mg/dL. INR: prothrombin time expressed as an international normalized ratio. AKI: Acute kidney injury. Mech. Vent.: Mechanical ventilation.

**Table 3 vaccines-12-00072-t003:** Main clinical data throughout the hospital stay and prescribed drugs.

Clinical		Vaccinated				*p* *	
Characteristic	No	ChAdOx1-S	BNT162b2	Inter-	No vs.	No vs.	ChAdOx1-S
	n = 288	n = 73	n = 101	Group	ChAdOx1-S	BNT162b2	vs. BNT162b2
Days hosp.							
All	8.9 ± 6.1	7.6 ± 5.1	8.4 ± 6.2	0.207			
D. alive	7.6 ± 5.1	6.8 ± 4.2	7.9 ± 5.1	0.492			
D. dead	10.2 ± 6.6	8.8 ± 6.0	9.2 ± 8.1	0.533			
Deaths	52.6%	41.1%	31.7%	0.001	0.052	<0.001	0.132
Death according to age							
<60 years	40.3%	29.0%	20.0%	0.245			
≥60 years	61.8%	50.0%	33.0%	<0.001	0.112	<0.001	0.047
Deaths according to Disease phase at admission							
Viral	34.2%	20.5%	19.1%	0.104			
Advanced **	59.8%	72.4%	43.4%	0.024	0.134	0.023	0.010
Deaths according to PSI at admission							
PSI score < 120	32.2%	21.7%	11.1%	0.002	0.115	0.001	0.107
PSI score ≥ 120	85.8%	76.0%	67.6%	0.046	0.180	0.016	0.336
Death with booster vaccination		38.1%	30.4%	0.414			

Percentages or averages and standard deviation are shown. * To compare three groups (intergroup analysis), ANOVA or likelihood ratio chi-square test was performed, to compare two groups, independent Student’s *t*-test or Fisher’s exact test was used, as appropriate for numerical or qualitative data, respectively. Days hosp.: Days of hospital stay. D. alive, and D. dead: Days of hospital stay in patients who lived and in patients who died, respectively. ** Advanced disease phase at admission: Pulmonary and hyper-inflammatory phases. PSI: Pneumonia Severity Index.

**Table 4 vaccines-12-00072-t004:** Relative risk from multivariate generalized linear mixed model with binary logistic regression link of various clinical characteristics and vaccination schedule to have a fatal outcome in patients hospitalized for COVID-19.

		95% CI		
Covariate	Ad RR	Lower	Upper	*p*
Age ≥ 60 years	1.46	0.91	2.37	0.119
Male	0.97	0.63	1.50	0.887
Diabetes	1.18	0.76	1.82	0.471
HBP	0.83	0.53	1.29	0.405
Smoking	4.50	2.00	10.14	<0.001
CKD	1.42	0.72	2.83	0.314
Charlson Index	2.67	0.60	11.97	0.199
Admission phase	4.01	2.47	6.50	<0.001
NEWS-2 score ≥ 12	3.56	1.64	7.76	0.001
PSI score > 120	8.14	5.06	13.09	<0.001
Mech. Vent.	3.74	1.89	7.39	<0.001
Creatinine ≥ 4mg/dL	0.89	0.39	2.04	0.777
AKI	1.10	0.63	1.92	0.726
Hemodialysis	0.70	0.16	3.09	0.633
Neutrophils ≥ 8 × 10e^3^/uL	3.76	2.34	6.02	<0.001
Lymphocytes ≥ 680/uL	0.53	0.34	0.84	0.007
AST ≥ 50UL	1.40	0.81	2.42	0.229
ALT ≥ 45UL	0.45	0.26	0.79	0.005
Ferritin ≥ 810 ng/mL	0.91	0.58	1.42	0.668
ESR ≥ 30 mm/h	1.42	0.66	3.05	0.370
**Vaccine type**				
BNT162b2/ChAdOx1-S 2 doses α	0.54	0.30	0.97	0.041
BNT162b2 2 doses α	0.41	0.22	0.79	0.008
ChAdOx1-S 2 doses α	1.04	0.48	2.29	0.915
Booster vaccination Ω	1.74	0.68	4.45	0.247

The multivariate statistical model included all the listed characteristics (fixed effects), which were considered as risk factors for mortality, their absence being classified as the reference, except for the type of vaccine administered, where the reference was the unvaccinated. Adjustment is for all variables listed, including vaccine type classified as; No, BNT162b2 (two doses), or ChAdOx1-S (two doses). The month of hospitalization (pandemic period) was included in the model as a random effect. HBP: Systemic arterial hypertension. Smoking: current smoker. CKD: Chronic kidney disease. Admission phase: Advanced disease phase at admission (Pulmonary and hyper-inflammatory phases, reference: viral phase). NEWS-2: National Early Warning Score 2. PSI: Pneumonia Severity Index. Mech. Vent.: Mechanical ventilation. AKI: Acute kidney injury. AST: aspartate amino transferase, expressed in international units per liter (IU/L). ALT: alanine aminotransferase, expressed in IU/L. ESR: Erythrocyte Sedimentation Rate, mm/hour. ChAdOx1-S: Oxford–AstraZeneca. BNT162b2: Pfizer–BioNTech. α In comparison with not vaccinated group. Ω Booster vaccination: only ChAdOx1-S (Oxford–AstraZeneca) was used. The dichotomization of variables was determined through an analysis with the area under the receiver operating characteristic (ROC) curve, with its respective cut-off point, calculated to discriminate subjects who died.

**Table 5 vaccines-12-00072-t005:** Relative risk from multivariate generalized linear mixed model with binary logistic regression link, comparing various clinical characteristics and the BNT162b2 vaccination schedule with ChAdOx1-S in patients hospitalized for COVID-19.

		95% CI		
Covariate	Ad RR	Lower	Upper	*p*
Age ≥ 60 years	7.61	2.74	21.11	<0.001
Male	1.42	0.62	3.24	0.407
Smoking	0.88	0.20	3.88	0.861
CKD	0.75	0.28	2.00	0.570
Charlson Index	1.07	0.82	1.39	0.633
Admission phase	2.30	1.14	4.63	0.020
NEWS-2 score ≥ 12	1.84	0.55	6.20	0.322
PSI score ≥ 120	3.91	1.50	10.22	0.006
Mech. Vent.	1.24	0.41	3.78	0.701
Creatinine ≥ 4mg/dL	0.58	0.17	2.00	0.389
AKI	0.88	0.37	2.06	0.762
Neutrophils ≥ 8 × 10e^3^/uL	0.87	0.37	2.05	0.753
Lymphocytes ≥ 680/uL	0.95	0.44	2.03	0.893
AST ≥ 50UL	1.42	0.61	3.30	0.410
ALT ≥ 45UL	0.46	0.17	1.24	0.123
Ferritin ≥ 810 ng/mL	0.21	0.09	0.46	<0.001
ESR ≥ 30 mm/h	0.91	0.58	1.42	0.668
Booster vaccination Ω	0.32	0.13	0.75	0.009
Death	0.27	0.10	0.70	0.008

The multivariate statistical model included and adjusted for all the listed characteristics (introduced into the model as fixed effects). The month of hospitalization (pandemic period) was included in the model as a random effect. Smoking: current smoker. CKD: Chronic kidney disease. Admission phase: advanced disease phase at admission (pulmonary and hyper-inflammatory phases, reference: viral phase). NEWS-2: National Early Warning Score 2. PSI: Pneumonia Severity Index. Mech. Vent.: Mechanical ventilation. AKI: Acute kidney injury. AST: aspartate amino transferase, expressed in international units per liter (IU/L). ALT: alanine aminotransferase, expressed in IU/L. ESR: Erythrocyte Sedimentation Rate, mm/hour. ChAdOx1-S: Oxford–AstraZeneca. BNT162b2: BNT162b2–BioNTech. Ω Booster vaccination: only ChAdOx1-S (Oxford–AstraZeneca) was used. The dichotomization of variables was determined through an analysis with the area under the receiver operating characteristic (ROC) curve, with its respective cut-off point, calculated to discriminate subjects who died.

**Table 6 vaccines-12-00072-t006:** Relative risk from multivariate generalized linear mixed model with binary logistic regression link of various clinical characteristics and BNT162b2 vaccination schedule in comparison with non-vaccinated individuals in patients hospitalized for COVID-19.

		95% CI		
Covariate	Ad RR	Lower	Upper	*p*
Age ≥ 60 years	3.25	1.70	6.22	0.000
Male	1.78	1.10	2.88	0.018
Smoking	0.38	0.15	0.94	0.037
CKD	0.69	0.35	1.37	0.297
Charlson Index	1.41	1.20	1.65	0.000
Admission phase	0.88	0.55	1.42	0.622
NEWS-2 score ≥ 12	1.18	0.62	2.26	0.603
PSI score ≥ 120	1.50	0.82	2.74	0.186
Mech. Vent.	1.13	0.60	2.12	0.686
Creatinine ≥ 4mg/dL	1.58	0.73	3.38	0.237
AKI	0.54	0.31	0.93	0.028
Neutrophils ≥ 8 × 10e^3^/uL	2.43	1.51	3.89	0.000
Lymphocytes ≥ 680/uL	3.63	2.34	5.62	0.000
AST ≥ 50UL	1.02	0.59	1.78	0.919
ALT ≥ 45UL	0.64	0.35	1.19	0.163
Ferritin ≥ 810 ng/mL	0.74	0.46	1.19	0.220
ESR ≥ 30 mm/h	1.03	0.47	2.22	0.939
Death	0.38	0.19	0.72	0.004

The multivariate statistical model included and adjusted for all the listed characteristics (introduced into the model as fixed effects). The month of hospitalization (pandemic period) was included in the model as a random effect. Smoking: current smoker. CKD: Chronic kidney disease. Admission phase: advanced disease phase at admission (pulmonary and hyper-inflammatory phases, reference: viral phase). NEWS-2: National Early Warning Score 2. PSI: Pneumonia Severity Index. Mech. Vent.: Mechanical ventilation. AKI: Acute kidney injury. AST: aspartate amino transferase, expressed in international units per liter (IU/L). ALT: alanine aminotransferase, expressed in IU/L. ESR: Erythrocyte Sedimentation Rate, mm/hour. ChAdOx1-S: Oxford–AstraZeneca. BNT162b2: Pfizer–BioNTech. Ω Booster vaccination: only ChAdOx1-S (Oxford–AstraZeneca) was used. The dichotomization of variables was determined through an analysis with the area under the receiver operating characteristic (ROC) curve, with its respective cut-off point, calculated to discriminate subjects who died.

**Table 7 vaccines-12-00072-t007:** Mean and *p*-values resulting from the analysis of repeated measurements during the first 6 days of hospitalization of some clinical parameters using multivariate linear mixed effects models.

	Mean, First 6 Days			*p*-Value No vs.		ChAdOx1-S vs.
	No	ChAdOx1-S	BNT162b2	ChAdOx1-S	BNT162b2	BNT162b2
Ferritin	1043 ± 758	1031 ± 813	632 ± 473	0.646	<0.001	<0.001
Lymphocytes	740.7 ± 776	1055 ± 699	1008 ± 836	<0.001	<0.001	0.148
PLR	1259 ± 2317	339 ± 301	423 ± 460	<0.001	<0.001	0.055
NEWS-2	7.8 ± 3.8	6.3 ± 3.5	6.1 ± 3.4	0.004	0.028	0.450
PSI score	110.5 ± 44	110.6 ± 39	119.3 ± 38	0.302	<0.001	<0.001

Mean ± standard deviation of the values during the first 6 days of hospitalization. *p*-values obtained by univariate linear mixed effects model tests. ChAdOx1-S: Oxford–AstraZeneca. BNT162b2: Pfizer–BioNTech. PSI: Pneumonia Severity Index. NEWS-2: National Early Warning Score. Ferritin: Serum values expressed in ng/mL. Lymphocytes: Blood values expressed in cells per microliter of blood. PLR: Platelet-to-lymphocyte ratio.

**Table 8 vaccines-12-00072-t008:** Cutoff scores, area under the curve (AUC), sensitivity, and specificity for various clinical measures examined in this study.

Variable	Group	AUC	S.E.	*p*	95% CI		Cutoff	Sens	Spec
PSI	Not vac	0.92	0.01	<0.001	0.91	0.93	118.0	0.80	0.10
	ChAdOx1-S	0.89	0.01	<0.001	0.86	0.92	123.5	0.78	0.17
	BNT162b2	0.85	0.02	<0.001	0.82	0.88	124.5	0.81	0.22
Oxemia	Not vac	0.35	0.02	<0.001	0.31	0.39	74.0	0.92	0.95
	ChAdOx1-S	0.35	0.04	0.001	0.26	0.43	84.5	0.83	0.93
	BNT162b2	0.41	0.03	0.007	0.34	0.47	82.5	0.83	0.86
Lymphocytes	Not vac	0.39	0.02	<0.001	0.36	0.42	535.0	0.44	0.59
	ChAdOx1-S	0.32	0.03	<0.001	0.27	0.37	831.4	0.37	0.67
	BNT162b2	0.38	0.03	<0.001	0.34	0.43	893.5	0.40	0.57
Neutrophils	Not vac	0.69	0.01	<0.001	0.67	0.72	7311.5	0.62	0.30
	ChAdOx1-S	0.62	0.03	<0.001	0.57	0.68	9710.9	0.65	0.43
	BNT162b2	0.72	0.02	<0.001	0.68	0.76	9430.1	0.70	0.37
LDH	Not vac	0.75	0.02	<0.001	0.72	0.78	367.5	0.62	0.25
	ChAdOx1-S	0.87	0.03	<0.001	0.82	0.92	332.7	0.79	0.17
	BNT162b2	0.76	0.03	<0.001	0.70	0.81	325.4	0.69	0.30
Ferritin	Not vac	0.64	0.02	<0.001	0.60	0.67	875.1	0.59	0.41
	ChAdOx1-S	0.57	0.04	0.090	0.49	0.64	878.4	0.53	0.46
	BNT162b2	0.67	0.03	<0.001	0.61	0.73	648.3	0.58	0.41
CRP	Not vac	0.74	0.02	<0.001	0.71	0.78	12.2	0.60	0.25
	ChAdOx1-S	0.57	0.05	0.162	0.47	0.67	12.3	0.56	0.49
	BNT162b2	0.71	0.01	<0.001	0.68	0.74	12.2	0.60	0.30
D dimer	Not vac	0.74	0.02	<0.001	0.70	0.77	1009.5	0.64	0.31
	ChAdOx1-S	0.71	0.04	<0.001	0.64	0.79	1231.7	0.62	0.48
	BNT162b2	0.69	0.03	<0.001	0.62	0.75	1083.0	0.63	0.40
eGFR	Not vac	0.32	0.01	<0.001	0.29	0.34	75.2	0.42	0.73
	ChAdOx1-S	0.43	0.03	0.009	0.37	0.48	36.3	0.46	0.59
	BNT162b2	0.431	0.024	0.005	0.384	0.479	60.00	0.436	0.598

AUC: Area under the curve. S.E.: Standard error. CI: Confidence intervals. Sens: Sensitivity. Spec: Specificity. Not vac: Not vaccinated. ChAdOx1-S: Oxford–AstraZeneca. BNT162b2: Pfizer–BioNTech. PSI: Pneumonia Severity Index. Oxemia: Arterial oxemia (%). Lymphocytes and neutrophils expressed in cells per microliter of blood. LDH: Lactate Dehydrogenase expressed in international units per liter. Ferritin expressed in ng/mL. CRP: C-Reactive Protein mg/dL. D dimer expressed in ng/mL. eGFR: estimated glomerular filtration rate, expressed in mL/min/1.73 m^2^.

**Table 9 vaccines-12-00072-t009:** Mean values of various clinical parameters throughout the entire hospitalization period categorized by vaccine type, along with *p*-values resulting from the analysis using multivariate linear mixed effects models.

	Vaccine			*p*-Value No vs.		ChAdOx1-S vs.
Variable	No	ChAdOx1-S	BNT162b2	ChAdOx1-S	BNT162b2	BNT162b2
PSI	119.1 ± 45.5	117.5 ± 41.1	128.1 ± 42.3	0.496	<0.001	<0.001
Oxemia	90.68 ± 10.18	90.91 ± 9.95	90.78 ± 10.43	0.826	0.925	0.807
pH	7.339 ± 0.16	7.333 ± 0.15	7.360 ± 0.14	0.314	0.034	0.014
Lymphocytes	691.9 ± 722.0	1040.4 ± 721.8	1091.1 ± 856.6	<0.001	<0.001	0.102
Neutrophils	8309.4 ± 6674.1	11,972.3 ± 7959.6	10,479.9 ± 5828.0	<0.001	<0.001	0.024
LDH	421.9 ± 400.2	416.3 ± 568.0	370.5 ± 258.2	0.814	0.046	0.166
ALP	92.4 ± 61.5	170.3 ± 162.8	106.1 ± 75.6	<0.001	0.006	0.024
AST	56.4 ± 177.4	80.6 ± 224.9	62.0 ± 171.5	0.148	0.628	0.402
ALT	45.2 ± 58.6	63.6 ± 149.6	49.2 ± 121.7	0.002	0.389	0.520
Ferritin	1129.4 ± 769.2	1130.3 ± 790.9	807.8 ± 616.3	0.673	<0.001	<0.001
Platelets	257.5 ± 136.5	264.5 ± 155.6	270.1 ± 131.3	0.057	0.894	0.130
MPV	10.7 ± 1.1	10.5 ± 1.0	10.7 ± 1.2	0.006	0.706	0.042
CRP	14.8 ± 19.9	14.1 ± 9.1	13.0 ± 17.7	0.699	0.230	0.562
D dimer	2475.5 + 12,065.1	3312.1 ± 4438.8	2090.2 ± 2737.1	0.444	0.611	0.002
eGFR	84.8 ± 66.2	62.8 ± 65.4	70.9 ± 53.8	0.004	< 0.001	0.826

No: not vaccinated. ChAdOx1-S: Oxford–AstraZeneca. BNT162b2: Pfizer–BioNTech. PSI: Pneumonia Severity Index. Oxemia: Arterial oxemia (%). Lymphocytes and neutrophils expressed in cells per microliter of blood. LDH: Lactate Dehydrogenase expressed in international units per liter (IU/L). ALP: alkaline phosphatase, expressed in IU/L. AST: aspartate amino transferase, expressed in IU/L. ALT: alanine aminotransferase, expressed in IU/L. Ferritin expressed in ng/mL. Platelets X1000, expressed in cells per microliter of blood. MPV: mean platelet volume expressed in femtoliter. CRP: C-Reactive Protein mg/dL. D dimer expressed in ng/mL. PLR: Platelet-to-lymphocyte ratio. eGFR: estimated glomerular filtration rate, expressed in mL/min/1.73 m^2^.

## Data Availability

The datasets used and/or analyzed during the current study are available from the corresponding author on reasonable request.

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
