# Peer review of "Differences in the Evolution of Clinical, Biochemical, and Hematological Indicators in Hospitalized Patients with COVID-19 According to Their Vaccination Scheme: A Cohort Study in One of the World’s Highest Hospital Mortality Populations"

_vaccines, 2024, doi:10.3390/vaccines12010072_

Round 1
Reviewer 1 Report
Comments and Suggestions for Authors
Very interesting data on how well vaccines protect against lethal Covid after hospitalization. It is clearly presented that the Pfizer vaccine offers better protection after hospitalization than the AstraZeneca vaccine. It would be interesting to couple this in the discussion with protection against hospitalization by both vaccines from other studies.
It is clear that vaccines protect against Covid and serious Covid (i.e., hospitalization) and the authors should raise this point both in their discussion and abstract to avoid confusion and abuse of the scientific conclusions for antivax purposes. This is also indicated by the increased health risks (HBP, CKD, Charlson index, >60 years) in the vaccinated people i.e., hospitalized vaccinated people were at greater risk than hospitalized unvaccinated people.
Please clarify “when stratifying by vaccine type … those vaccinated with AstraZeneca did not differ from the unvaccinated in terms of their risk of death (adjusted RR1.04)” (line 481-484). The value seems derived from Table IV, but the authors do not explain how the adjusted RR is calculated. Its value is at odds with other values in the paper, like Pfizer/Astra 2 doses only 0.54 (Table IVb) with 0.41 and 101/73 Pfizer to Astra. They are also at odds with data from Table IIa and III. Patients vaccinated with AstraZeneca group had in general a much poorer health (Table IIa) than unvaccinated patients. The average mortality rate was lower in the AstraZeneca group (53% versus 41%) (Table III). With better survival in patients with poorer prognosis, a positive effect was anticipated. Please clarify how the adjusted RR 1.04 and 1.74 are calculated for Astra 2 doses and additional dose omega, respectively. It is remarkable that increased risk factors (described above) seem to have a decreased effect on protection.
Comments on the Quality of English Language
Please check use of Capitals within sentences.
Author Response
Reviewer 1
COMMENT: Very interesting data on how well vaccines protect against lethal Covid after hospitalization. It is clearly presented that the Pfizer vaccine offers better protection after hospitalization than the AstraZeneca vaccine. It would be interesting to couple this in the discussion with protection against hospitalization by both vaccines from other studies.
It is clear that vaccines protect against Covid and serious Covid (i.e., hospitalization) and the authors should raise this point both in their discussion and abstract to avoid confusion and abuse of the scientific conclusions for antivax purposes. This is also indicated by the increased health risks (HBP, CKD, Charlson index, >60 years) in the vaccinated people i.e., hospitalized vaccinated people were at greater risk than hospitalized unvaccinated people.
RESPONSE: In response to your first and second inquiry. The abstract was subtly adjusted to underscore the importance of vaccination, and in the discussion section, a paragraph was incorporated emphasizing the significance of vaccination and drawing comparisons with other studies.
Abstract: COVID-19 vaccines primarily prevent severe illnesses or hospitalization, but there is limited data on their impact during hospitalization for seriously ill patients. In a Mexican cohort with high COVID-19 mortality, a study assessed vaccination's effects. From 2021 to 2022, 462 patients with 4,455 hospital days were analyzed. The generalized multivariate linear mixed model (GENLINMIXED) with binary logistic regression link, survival analysis and ROC curves were used to identify risk factors for death. The results showed that the vaccinated individuals were almost half as likely to die (adRR=0.54, 95% CI=0.30-0.97, P=0.041). When stratifying by vaccine, the BNT162b2 group had a 2.4 times lower risk of death (adRR=0.41, 95% CI=0.2-0.8, P=0.008), while the ChAdOx1-S group did not significantly differ from the non-vaccinated (adRR=1.04, 95% CI=0.5-2.3, P=0.915). The BNT162b2 group exhibited a higher survival, the unvaccinated showed increasing mortality, and the ChAdOx1-S group remained intermediate (P=0.003, multigroup log-rank test). Additionally, BNT162b2-vaccinated individuals had lower values for markers, such as ferritin and D-dimer. Biochemical and hematological indicators suggested a protective effect of both types of vaccines, possibly linked to higher lymphocyte counts and lower platelet-to-lymphocyte ratio (PLR). It is imperative to highlight that these results reinforce the efficacy of COVID-19 vaccines. However, further studies are warranted for a comprehensive understanding of these findings.
Discussion: Considering the results obtained in this study, it is imperative to highlight that these findings reinforce the efficacy of COVID-19 vaccines in reducing hospital fatalities, in addition to the previously reported reduction in severity and hospitalizations [3,4]. However, addressing these findings with caution is fundamental to avoid misinterpretations that could be misused for anti-vaccine purposes. It is crucial to emphasize that this study not only highlights a significant decrease in the risk of mortality for vaccinated individuals, especially in populations with increased health risks, but also underscores notable differences among the various types of vaccines used. The findings highlight the crucial and beneficial role of vaccination, adding to what was previously described to reduce symptoms and the risks of hospital admission and death [37,38]. These collective results underscore the current importance of vaccination efforts, considering variations in populations and the types of vaccines used.
- Mascellino, M.T.; Di Timoteo, F.; De Angelis, M.; Oliva, A. Overview of the Main Anti-SARS-CoV-2 Vaccines: Mechanism of Action, Efficacy and Safety. Infect Drug Resist 2021, Volume 14, 3459–3476, doi:10.2147/IDR.S315727.
- Havers, F.P.; Pham, H.; Taylor, C.A.; Whitaker, M.; Patel, K.; Anglin, O.; Kambhampati, A.K.; Milucky, J.; Zell, E.; Moline, H.L.; et al. COVID-19-Associated Hospitalizations Among Vaccinated and Unvaccinated Adults 18 Years or Older in 13 US States, January 2021 to April 2022. JAMA Intern Med 2022, 182, 1071, doi:10.1001/jamainternmed.2022.4299.
- Rosero-Bixby, L. The Effectiveness of Pfizer-BioNTech and Oxford-AstraZeneca Vaccines to Prevent Severe COVID-19 in Costa Rica: Nationwide, Ecological Study of Hospitalization Prevalence. JMIR Public Health Surveill 2022, 8, e35054, doi:10.2196/35054.
- Lopez Bernal, J.; Andrews, N.; Gower, C.; Robertson, C.; Stowe, J.; Tessier, E.; Simmons, R.; Cottrell, S.; Roberts, R.; O’Doherty, M.; et al. Effectiveness of the Pfizer-BioNTech and Oxford-AstraZeneca Vaccines on Covid-19 Related Symptoms, Hospital Admissions, and Mortality in Older Adults in England: Test Negative Case-Control Study. BMJ 2021, n1088, doi:10.1136/bmj.n1088.
COMMENT: Please clarify “when stratifying by vaccine type … those vaccinated with AstraZeneca did not differ from the unvaccinated in terms of their risk of death (adjusted RR1.04)” (line 481-484). The value seems derived from Table IV, but the authors do not explain how the adjusted RR is calculated. Its value is at odds with other values in the paper, like Pfizer/Astra 2 doses only 0.54 (Table IVb) with 0.41 and 101/73 Pfizer to Astra. They are also at odds with data from Table IIa and III. Patients vaccinated with AstraZeneca group had in general a much poorer health (Table IIa) than unvaccinated patients. The average mortality rate was lower in the AstraZeneca group (53% versus 41%) (Table III). With better survival in patients with poorer prognosis, a positive effect was anticipated. Please clarify how the adjusted RR 1.04 and 1.74 are calculated for Astra 2 doses and additional dose omega, respectively. It is remarkable that increased risk factors (described above) seem to have a decreased effect on protection.
RESPONSE: Thank you for the observation. Now, the text is clearer regarding the points you mentioned. The methodology for calculating the adjusted RR is now described in greater detail and with more references in the "Statistical Analysis" section, as follows:
Association analysis was conducted using multivariate generalized linear mixed models (GLMM, GENLINMIXED in SPSS) with a binary logistic regression link and separate random intercepts (SPRI), as previously described [21, 22]. Data were summarized as relative risks (RRs) with 95% confidence intervals (CIs) and P-values, adjusted for multiple variables. GLMM is a valid strategy for estimating RRs in multivariate analysis.
- Jaffa, M.A.; Gebregziabher, M.; Luttrell, D.K.; Luttrell, L.M.; Jaffa, A.A. Multivariate Generalized Linear Mixed Models with Random Intercepts to Analyze Cardiovascular Risk Markers in Type-1 Diabetic Patients. J Appl Stat 2016, 43, 1447–1464, doi:10.1080/02664763.2015.1103708.
- Naimi, A.I.; Whitcomb, B.W. Estimating Risk Ratios and Risk Differences Using Regression. Am J Epidemiol. 2020, 189, 508–510, doi:10.1093/aje/kwaa044.
Regarding the sentence: "those vaccinated with AstraZeneca did not differ from the unvaccinated in terms of their risk of death (adjusted RR1.04)" (lines 481-484). The value seems to be derived from Table IV, but the authors do not explain how the adjusted RR is calculated. The legend "(see Table IV)" has been added next to the data "(adjusted RR1.04)" to clarify the origin of the data in the text. How is the adjusted RR calculated? It is now explained in more detail in the "Statistical Analysis" section, while the title of Table IV specifies the method from which the RRs are derived: "Table IV. Relative risk from multivariate generalized linear mixed model with binary logistic regression link of various clinical characteristics and vaccination schedule to have a fatal outcome in patients hospitalized for COVID-19." Additionally, in the body text below Table IV (as well as in Tables V and VI), details about the statistical analysis have been included: in Table IV: The multivariate statistical model included all the listed characteristics (as fixed effects), which were considered as risk factors for mortality, with their absence classified as the reference, except for the type of vaccine administered, where the reference was the unvaccinated. Adjustment is made for all listed variables, including vaccine type classified as; No, Pfizer 2 doses, or Astra 2 doses. In Tables V and VI: The multivariate statistical model included and adjusted for all the listed characteristics (introduced into the model as fixed effects).
To address the observation about the discrepancy in values within the paper, particularly with respect to the mortality rate of patients vaccinated with AstraZeneca compared to other groups, the following response has been provided:
In the section describing Table IV, the text now includes the following addition: "Combining the patients vaccinated with BNT162b2 and ChAdOx1-S into a single group, it is observed that patients in this group reduce their risk of death by almost half, with an RR value of 0.54 (95% CI 0.30-0.97), being a congruent value as it is located between the RR values of Pfizer (0.41) and Astra (1.04)."
Regarding the perceived benefit of being vaccinated with AstraZeneca (ChAdOx1-S), the text has consistently highlighted a protective effect through various analyses, such as survival curves, mortality rates, and hematological parameters. The survival curves (Figure 2) and the mortality rate (Table III) indicate that individuals vaccinated with AstraZeneca exhibit intermediate values between unvaccinated patients and those vaccinated with Pfizer. This is explicitly stated in the CONCLUSION, where it is mentioned:
Patients vaccinated with ChAdOx1-S had an intermediate mortality rate between BNT162b2-vaccinated patients (who had the lowest mortality) and the unvaccinated group (with the highest mortality). The protective effect observed in vaccinated patients can be explained by higher lymphocyte counts and lower levels of some inflammation markers such as PLR. However, patients vaccinated with BNT162b2 additionally have lower values of other inflammation markers, such as ferritin and D-dimer.
Additionally, it is evident that the protective effect of vaccination is more pronounced in those vaccinated with Pfizer compared to the unvaccinated and those vaccinated with ChAdOx1-S, as indicated by the RR values in Tables IV and VI, along with a lower mortality rate in specific patient groups (Table III). Consequently, this more substantial protective effect with Pfizer is explicitly acknowledged in the conclusions of the text.
While the beneficial effect of ChAdOx1-S vaccination is apparent, it is acknowledged to be limited compared to Pfizer. The perceived inconsistency of the adjusted RR value of 1.0 for the ChAdOx1-S group is addressed by explaining that it results from a multivariate generalized linear mixed model with a binary logistic regression link, considering all adjustment variables listed in Table IV. The text emphasizes that differences in comorbidities, smoking, and phases of hospital admission were considered in the adjustment, leading to the reported adjusted RR.
To address the need for a comprehensive assessment, a sentence has been added in the discussion section:
Another relevant aspect is that the effect of vaccination was analyzed from different points of view (mortality rate, survival curves, association analysis, ROC curves, evaluation of hematological and biochemical parameters), so conclusions can be reached based on multiple aspects and not with the result of a single analysis.
Regarding the RR of the additional dose (booster), this value was not significant, only "The receipt of a booster dose was also not associated with the risk of death (P=0.247)" is mentioned. The adjustment to determine the association between “booster vaccination and death was under the same mechanism, explained in methods and in the lower text of the tables.
COMMENT: Please check use of Capitals within sentences.
RESPONSE: Thank you for your observation, the sentences were modified using Capitals according Journal recommendations.
Reviewer 2 Report
Comments and Suggestions for Authors
This manuscript delineates the hospitalised COVID-19 patients. The outcomes are very interesting, and the analyses are more and slightly deep in detail. The technical was sound. However, some points need to be revised to improve the quality of the manuscript.
Major concerns.
1. I suggest avoiding using the manufacturer name of the vaccine because the same manufacturer may produce more than one type of COVID-19 vaccine, such as inhalation route, lite" version or bivalent, XBB monovalent.
Using manufacturer's names could lead to confusion, especially in the future, when the reader may not recall the COVID-19 vaccine used during the crisis. Moreover, using the manufacturer's name is not a "professional" use.
Suggest using a research name or commercial name to make it clear which vaccines were used at that time.
Commercial name: Covilo, CoronaVac, Jcovden, Convidecia, Vaxzevria, Sputnik V, Nuvaxovid, Comirnaty, Spikevax, Spikevax bivalent Original/Omicron BA.1, Spikevax bivalent Original/Omicron BA.4-5.
Research name: BBIBP-CorV, CoronaVac, Ad26.COV2.S, AD5-nCoV, ChAdOx1-S, Gam-COVID-Vac, NVX-CoV2373, BNT162b2, mRNA-1273, mRNA-1273.214, mRNA-1273.222.
Did you see it? The second-generation (bivalent) vaccines were made from the same manufacturer. Suppose you use the manufacturer's name. The reader will be confused about which vaccine was used at that time.
I suggest using commercial names or research names throughout the manuscript, tables and figures. You may be using an abbreviation in the table to make it fit into the column space.
2. Lines 53-57. Suggest rewriting it like the example below.
1) mRNA-based vaccines such as BNT162b2 (Pfizer—BioNTech) and mRNA-1273 (Moderna—NIAID); 2) adenoviral vector-based vaccines like ChAdOx1-S (Oxford— AstraZeneca), Ad26.COV2.S (Johnson & Johnson), Gam-COVID-Vac (Gamaleya), and Ad5nCoV (CanSino); and 3) inactivated coronavirus-based vaccines like CorovaVac (Sinovac) and BBIBP-CorV (Sinopharm).
3. Did all participants get their first infection?
Re-infection may alter the outcomes due to remaining hybrid immunity, which could provide greater protection than vaccination alone.
4. Did some participants get a booster vaccination?
If it was, yes. How to analyse? Excluded or further analysed in the subgroup?
Booster vaccination recipients have a low proportion, but there could altered outcomes.
5. High BMI is one of the factors related to severe infection.
6. I suggest adding more information (a subgroup of the outcomes and discussion) about each prominent variant wave. A subgroup may be added to the supplementary materials if it works. It could be more informative about how each wave may affect morbidity and mortality because this study collected data covering pre-B.1.1.7 (Alpha), B.1.1.7 (Alpha), B.1.617.2 (Delta), and B.1.1.529 (Omicron), Each variant could give different clinical and laboratory characteristics that could make your paper more impact. You can see the period of each prominent variant wave in Mexico from the literature.
Comments.
1. Line 55 "ChAdOx1" is not COVID-19, it only viral vector. The "ChAdOx1-S" is a COVID-19 vaccine that embedded a spike gene of the SARS-CoV-2 to the viral vector platform.
Moreover, the ChAdOx1 platform has various vaccines, such as ChAdOx1 85A (tuberculosis), ChAdOx1 sCHIKV (Chikungunya), ChAdOx1-biEBOV (ebola) and so on, depending on the embedded antigen gene(s).
2. Were you tried to subgroup in the age group, such as 0-17, 18-35, 36-59 and 60+?
I think the age group may be more interesting to see the risk factors if it was work.
3. Table 1 suggests adding a parameter of ≥60 years old because almost hospitalised or severe infection occurs in senescence patients.
4. Table 1. What is an additional dose? Do you mean "booster vaccination"?
5. Why do you use the term "Evolution"?
It seems the virus could evolve to differentiate the antigenic in their patient's host or change the practice from old-fashioned to modernised.
Typos.
1. Line 57 "BIBP-CorV". Do you mean "BBIBP-CorV"?
Comments on the Quality of English LanguagePlease check typos that embedded a dash, such as line 57 "glob-al", and line 67 "var-ied".
Author Response
Reviewer 2
COMMENT: This manuscript delineates the hospitalized COVID-19 patients. The outcomes are very interesting, and the analyses are more and slightly deep in detail. The technical was sound. However, some points need to be revised to improve the quality of the manuscript.
Major concerns.
- I suggest avoiding using the manufacturer name of the vaccine because the same manufacturer may produce more than one type of COVID-19 vaccine, such as inhalation route, lite" version or bivalent, XBB monovalent.
Using manufacturer's names could lead to confusion, especially in the future, when the reader may not recall the COVID-19 vaccine used during the crisis. Moreover, using the manufacturer's name is not a "professional" use.
Suggest using a research name or commercial name to make it clear which vaccines were used at that time.
Commercial name: Covilo, CoronaVac, Jcovden, Convidecia, Vaxzevria, Sputnik V, Nuvaxovid, Comirnaty, Spikevax, Spikevax bivalent Original/Omicron BA.1, Spikevax bivalent Original/Omicron BA.4-5.
Research name: BBIBP-CorV, CoronaVac, Ad26.COV2.S, AD5-nCoV, ChAdOx1-S, Gam-COVID-Vac, NVX-CoV2373, BNT162b2, mRNA-1273, mRNA-1273.214, mRNA-1273.222.
Did you see it? The second-generation (bivalent) vaccines were made from the same manufacturer. Suppose you use the manufacturer's name. The reader will be confused about which vaccine was used at that time.
I suggest using commercial names or research names throughout the manuscript, tables and figures. You may be using an abbreviation in the table to make it fit into the column space.
RESPONSE: Thank you for your feedback. In line with your suggestion, we have revised the manuscript by replacing manufacturer names with research names throughout. For instance, we now use "BNT162b2" for the Pfizer–BioNTech vaccine and "ChAdOx1-S" for the Oxford–AstraZeneca vaccine. We believe these changes contribute to a clearer and more professional presentation. Your input has been immensely helpful, and we appreciate your ongoing support.
- Lines 53-57. Suggest rewriting it like the example below.
1) mRNA-based vaccines such as BNT162b2 (Pfizer—BioNTech) and mRNA-1273 (Moderna—NIAID); 2) adenoviral vector-based vaccines like ChAdOx1-S (Oxford— AstraZeneca), Ad26.COV2.S (Johnson & Johnson), Gam-COVID-Vac (Gamaleya), and Ad5nCoV (CanSino); and 3) inactivated coronavirus-based vaccines like CorovaVac (Sinovac) and BBIBP-CorV (Sinopharm).
RESPONSE: Thank you for your input. We have revised lines 53-57 based on your suggestion. The updated text now reads as follows:
Mentioned vaccines include: 1) mRNA-based vaccines such as BNT162b2 (Pfizer—BioNTech) and mRNA-1273 (Moderna—NIAID); 2) adenoviral vector-based vaccines like ChAdOx1-S (Oxford—AstraZeneca), Ad26.COV2.S (Johnson & Johnson), Gam-COVID-Vac (Gamaleya), and Ad5nCoV (CanSino); and 3) inactivated coronavirus-based vaccines like CorovaVac (Sinovac) and BBIBP-CorV (Sinopharm).
COMMENT 3: Did all participants get their first infection?
Re-infection may alter the outcomes due to remaining hybrid immunity, which could provide greater protection than vaccination alone.
RESPONSE: Thank you for your observation. At the end of the first paragraph of the Results section (3.1 Patient Characteristics), information about previous infections was included: "Only 2 (0.43%) of these 462 patients reported having had a previous COVID-19 infection, so this characteristic was not included as a variable in the analyses.
COMMENT 4: Did some participants get a booster vaccination?
If it was, yes. How to analyse? Excluded or further analysed in the subgroup?
Booster vaccination recipients have a low proportion, but there could altered outcomes.
RESPONSE: The booster vaccination was analyzed using adjusted Relative Risk (RR) in Table IV, which does not indicate a significant modification of the risk of death. This is evident in the "3.4 Risk Factors for Death in Hospitalized Patients with COVID-19, Including Vaccination Schedule" section, where it is stated: "The receipt of a booster dose was also not associated with the risk of death (P=0.247) (see Table IV).
Additionally, in the discussion section, in the paragraph addressing the study's limitations, the following was added: "Likewise, only 8.9% of the hospitalized patients included had a booster vaccine, so a detailed analysis of this subgroup was not possible and is desirable in future research.
COMMENT 5: High BMI is one of the factors related to severe infection.
RESPONSE: Response: Thank you for your observation. In Table II, the BMI value was included for each group (not vaccinated and vaccinated with ChAdOx1-S or BNT162b2), along with the corresponding P value resulting from their comparison, which was P=0.264. Similarly, in the paragraph of section "3.2 Differences in Clinical Characteristics Between Vaccinated and Unvaccinated Individuals," the following sentence was added:
The body mass index (BMI) was not different between the 3 groups (P=0.264, ANOVA test) (table II), so it would not be a factor that could differentiate the clinical evolution between them.
Comment 6: I suggest adding more information (a subgroup of the outcomes and discussion) about each prominent variant wave. A subgroup may be added to the supplementary materials if it works. It could be more informative about how each wave may affect morbidity and mortality because this study collected data covering pre-B.1.1.7 (Alpha), B.1.1.7 (Alpha), B.1.617.2 (Delta), and B.1.1.529 (Omicron), Each variant could give different clinical and laboratory characteristics that could make your paper more impact. You can see the period of each prominent variant wave in Mexico from the literature.
RESPONSE: Your observation is very interesting. A figure illustrating variations in disease severity among different patient groups based on the pandemic period (with predominant variants in each period) has been added. Variations in disease severity can be observed across different periods among vaccinated patients, while in the non-vaccinated, disease severity remained similar in all periods. Additionally, information from the statistical model has been added to the association tables to specify the following: “The month of hospitalization (pandemic period) was included in the model as a random effect”. An association analysis for each period would be very interesting, but it would make the manuscript excessively lengthy. Therefore, in the discussion section, in the limitation’s subsection, it was added that this would be an analysis for future research. However, the added information on this topic, regarding severity in different periods among diverse groups, strengthens the manuscript.
Figure 2. Pneumonia Severity Index of patients with different vaccination schemes according to the pandemic period. It is observed that vaccinated patients show variations in severity depending on the pandemic period, while the severity of the non-vaccinated remained constant. In the period when the Delta variant was predominant, vaccinated patients experienced higher severity than the non-vaccinated, exhibiting the opposite trend in other periods. Periods were established based on the predominant variants in the general population of Mexico ([25–27]; January-July 2021: B.1.1.222; B.1.1.519; and Gamma. August-December 2021: Delta. January-April 2022: Delta, Omicron BA.1; Omicron BA.1.1, Omicron BA.1.15. May-December 2022: Omicron, others. *P<0.01. Mean +95% CI is shown.
COMMENTS 7: Line 55 "ChAdOx1" is not COVID-19, it only viral vector. The "ChAdOx1-S" is a COVID-19 vaccine that embedded a spike gene of the SARS-CoV-2 to the viral vector platform.
Moreover, the ChAdOx1 platform has various vaccines, such as ChAdOx1 85A (tuberculosis), ChAdOx1 sCHIKV (Chikungunya), ChAdOx1-biEBOV (ebola) and so on, depending on the embedded antigen gene(s).
RESPONSE: Thank you for bringing this to our attention. We appreciate your clarification. We have updated line 55 to reflect the correct information. The revised text now accurately designates "ChAdOx1-S" as a COVID-19 vaccine that incorporates the spike gene of the SARS-CoV-2 into the viral vector platform.
COMMENT 8: Were you tried to subgroup in the age group, such as 0-17, 18-35, 36-59 and 60+?
I think the age group may be more interesting to see the risk factors if it was work.
RESPONSE: Thank you for your observation. The patient groups were stratified as ≥60 years old or younger, as this cutoff has been identified as a factor influencing the disease outcome. Other age subgroups were not analyzed, primarily due to the fact that more than half of the patients were over 60 years old, and in the BNT162b2 group, they comprised the vast majority. In the limitations section of the discussion, it is mentioned that future research is necessary in additional patient subgroups, considering their age and with fewer comorbidities.
COMMENT 9: Table 1 suggests adding a parameter of ≥60 years old because almost hospitalized or severe infection occurs in senescence patients.
RESPONSE: Thank you for the observation. The suggested data has been added to Table I.
COMMENT 10: Table 1. What is an additional dose? Do you mean "booster vaccination"?
RESPONSE: Thank you for your observation. We have taken note of your suggestion, and the term "booster vaccination" has been incorporated to better reflect the administration of additional doses. We appreciate your attention to detail and your valuable input.
COMMENT 11: Why do you use the term "Evolution"?
It seems the virus could evolve to differentiate the antigenic in their patient's host or change the practice from old-fashioned to modernised.
RESPONSE: Thank you for your question. Given that the context of "evolution" in our text is referring to changes in clinical, biochemical, and hematological indicators in hospitalized patients with COVID-19, it seems appropriate to maintain the term "evolution" if it accurately conveys the intended meaning. In this context, "evolution" likely describes the progression or changes in the health status or conditions of patients over time.
COMMENT TYPOS: Line 57 "BIBP-CorV". Do you mean "BBIBP-CorV"?
RESPONSE: Thank you for bringing this to our attention. Yes, you are correct; it should be "BBIBP-CorV" instead of "BIBP-CorV" in line 57. Additionally, we appreciate your feedback on the embedded dash typos in lines 57 and 67. Rest assured, all identified errors have been rectified, and the necessary changes have been made to improve the overall quality of the manuscript.
COMMENT: Please check typos that embedded a dash, such as line 57 "glob-al", and line 67 "var-ied".
RESPONSE: Thank you for pointing out those typos. We have carefully reviewed the manuscript and made the necessary corrections, including removing the embedded dashes in words like "global" (line 57) and "varied" (line 67). These adjustments have been implemented to ensure the accuracy and clarity of the text. Your attention to detail is greatly appreciated.

Round 2
Reviewer 2 Report
Comments and Suggestions for Authors
Thank you for your revision to improve your manuscript quality and information. A large sample size could be informative, especially in each prominent variant wave.
I think it could be an error on your file because there seems no strikethrough or any deletion on the vaccine name throughout the manuscript, such as line 185 "ChAdOx1-SAstraZeneca or BNT162b2 (Pfizer—BioNTech) Pfizer–BioNTech"
Please check it again.
Comments and Typos.
1. Line 33, suggests using the "Pfizer group (BNT162b2)".
2. Line 34, suggests using the "AstraZeneca group (ChAdOx1-S)".
3. Lines 35, and 37 you already mentioned, which "Pfizer" used in this study in line 33. You can use only "Pfizer group" without (BNT162b2). Otherwise, you can use BNT162b2.
4. Line 36, you already mentioned, which "AstraZeneca" used in this study in line 34. You can use only "AstraZeneca group" without (ChAdOx1-S). Otherwise, you can use ChAdOx1-S.
5. Line 61 suggests using "(ChAdOx1-S)".
6. Line 141 suggests using "(with 2 doses of BNT162b2 (Pfizer—BioNTech) or ChAdOx1-S (AstraZeneca) vaccines)".
7. After lines 181-182, you can use only the vaccine's name without mentioning the manufacturer's name because you already first mentioned it in lines 181-182.
8. Figure 2. You may add a pattern to make each subgroup different because some readers may print it out as greyscale (Black and White).
You can test by using your smartphone camera with a monochrome function to make sure that is greyscale-friendly.
Suggest checking typos and parentheses.
Author Response
Reviewer 2
COMMENT: Thank you for your revision to improve your manuscript quality and information. A large sample size could be informative, especially in each prominent variant wave.
I think it could be an error on your file because there seems no strikethrough or any deletion on the vaccine name throughout the manuscript, such as line 185 "ChAdOx1-SAstraZeneca or BNT162b2 (Pfizer—BioNTech) Pfizer–BioNTech"
Please check it again.
RESPONSE: Thank you for your feedback on the manuscript, and I appreciate your diligence in reviewing the content. I've carefully addressed your concerns and made the necessary revisions. In response to your first inquiry in the discussion section we added a limitation of the present study:
The present study also had limitations. The number of patients did not allow for evaluations and comparisons between patients with a single vaccine dose. Similarly, the mortality in patients vaccinated with ChAdOx1-S was intermediate between the unvaccinated and those vaccinated with BNT162b2, so in various analyses, statistically significant differences between ChAdOx1-S and the other two groups were not observed. Future analyses with a larger number of individuals are desirable to better observe the differences between these groups. Additional facets of the investigation pertinent to the studied population, such as the types of vaccines administered and who was given priority for vaccination at the beginning of the pandemic in Mexico, did not allow for the evaluation of other vaccine brands and types, as well as, their effects on various population strata with fewer comorbidities.
In respect to the error on our file; It's possible that the absence of strikethrough or deletion markings on the vaccine names in the manuscript, such as in line 185 ("ChAdOx1-SAstraZeneca or BNT162b2 (Pfizer—BioNTech) Pfizer–BioNTech"), was due to an error in the attachment of the Track Changes version in PDF format.
To ensure clarity, I have rectified this issue and am resubmitting the revised manuscript with only the modifications highlighted. I apologize for any confusion, and I appreciate your understanding.
Please find the revised manuscript attached. If you have any further comments or require additional clarification, please do not hesitate to let me know.
Thank you for your time and attention to this matter.
COMMENTS AND TYPOS.
COMMENT 1. Line 33, suggests using the "Pfizer group (BNT162b2)".
RESPONSE: Thank you for your detailed feedback. We have carefully considered your suggestion and have made the recommended changes. In line with your comment, we have now updated Line 33 to read as follows: "Pfizer group (BNT162b2)." (Highlighted in yellow)
COMMENT 2. Line 34, suggests using the "AstraZeneca group (ChAdOx1-S)".
RESPONSE: Thank you for your insightful feedback. We have incorporated your suggestion into the revised manuscript. Line 34 now reads: "AstraZeneca group (ChAdOx1-S)." (Highlighted in yellow)
COMMENT 3. Lines 35, and 37 you already mentioned, which "Pfizer" used in this study in line 33. You can use only "Pfizer group" without (BNT162b2). Otherwise, you can use BNT162b2.
RESPONSE: Thank you for your continued attention to detail. We appreciate your guidance. We have taken your suggestion into account, and the necessary changes have been implemented. In Lines 35 and 37, we now refer to the "Pfizer group" without specifying (BNT162b2). (Highlighted in yellow)
COMMENT 4. Line 36, you already mentioned, which "AstraZeneca" used in this study in line 34. You can use only "AstraZeneca group" without (ChAdOx1-S). Otherwise, you can use ChAdOx1-S.
RESPONSE: Thank you once again for your diligence in reviewing our manuscript. We have carefully considered your suggestion, and the appropriate changes have been made. In Line 36, we now refer to the "AstraZeneca group" without specifying (ChAdOx1-S). (Highlighted in yellow)
COMMENT 5. Line 61 suggests using "(ChAdOx1-S)".
RESPONSE: Thank you for your ongoing review. We have incorporated your suggestion, and Line 61 now includes "(ChAdOx1-S)" as per your recommendation.
COMMENT 6. Line 141 suggests using "(with 2 doses of BNT162b2 (Pfizer—BioNTech) or ChAdOx1-S (AstraZeneca) vaccines)".
RESPONSE: Thank you for your continued review and valuable suggestions. We have implemented the change in Line 141, which now reads "[with 2 doses of BNT162b2 (Pfizer—BioNTech) or ChAdOx1-S (AstraZeneca) vaccines]" as you recommended. (Highlighted in yellow)
COMMENT 7. After lines 181-182, you can use only the vaccine's name without mentioning the manufacturer's name because you already first mentioned it in lines 181-182.
RESPONSE: Thank you for your observation. We have revised the text following Lines 181-182 accordingly. Moving forward, we now refer to the vaccine by name without reiterating the manufacturer, as initially stated in those lines.
COMMENT 8. Figure 2. You may add a pattern to make each subgroup different because some readers may print it out as greyscale (Black and White).
You can test by using your smartphone camera with a monochrome function to make sure that is greyscale-friendly.
RESPONSE: Thank you for your suggestions and additional comments. We have made modifications to Figure 2 taking into consideration your recommendation to add distinctive patterns to each subgroup for improved readability in greyscale.
To ensure greyscale-friendliness, we conducted tests using the monochrome function on a smartphone camera, as you suggested.
